# Endogenous retroviruses and TDP-43 proteinopathy form a sustaining feedback driving intercellular spread of *Drosophila* neurodegeneration

Yung-Heng Chang [1] & Josh Dubnau [1,2]

Inter-cellular movement of "prion-like" proteins is thought to explain propagation of neurodegeneration between cells. For example, propagation of abnormally phosphorylated cytoplasmic inclusions of TAR-DNA-Binding protein (TDP-43) is proposed to underlie progression of amyotrophic lateral sclerosis (ALS) and frontotemporal dementia (FTD). But unlike transmissible prion diseases, ALS and FTD are not infectious and injection of aggregated TDP-43 is not sufficient to cause disease. This suggests a missing component of a positive feedback necessary to sustain disease progression. We demonstrate that endogenous retrovirus (ERV) expression and TDP-43 proteinopathy are mutually reinforcing. Expression of either *Drosophila* mdg4-ERV (gypsy) or the human ERV, HERV-K (HML-2) are each sufficient to stimulate cytoplasmic aggregation of human TDP-43. Viral ERV transmission also triggers TDP-43 pathology in recipient cells that express physiological levels of TDP-43, whether they are in contact or at a distance. This mechanism potentially underlies the TDP-43 proteinopathy-caused neurodegenerative propagation through neuronal tissue.

Cytoplasmic aggregation of abnormally phosphorylated TDP-43 (pTDP-43) is seen in both neurons and glia in the affected brain regions of 97% of patients with ALS, ~40% of cases with FTD, in many cases of Alzheimer's disease (AD), and in patients with an AD-like presentation (Limbic-predominant Age-related TDP-43 Encephalopathy)[1–3]. Mutations in TDP-43 are responsible for a fraction of the inherited forms of ALS, providing a direct causal link between this gene and neurodegenerative cascades. But the vast majority of cases involve the mis-localization and aggregation of TDP-43 of normal amino acid sequence[1]. Thus, initiation of TDP-43 protein pathology is triggered by upstream factors, likely ones that accumulate with age. On the other hand, disease progression after diagnosis is relatively rapid, and is thought to involve inter-cellular propagation of disease pathology from a point of stochastic focal origin. This inflection in rate of disease progression after decades of apparently normal physiology is consistent with involvement of a self-sustaining biological processes, a positive feedback loop[1–6].

One model that has been proposed to explain such a self-sustaining inter-cellular propagation is the prion hypothesis[3–6]. In this model, pathological forms of misfolded TDP-43 are able to template misfolding in trans of normal wild-type TDP-43, much as has been demonstrated for the prion protein (PrP) that underlies infectious prion diseases[7,8]. Such misfolded TDP-43 is proposed to be released from neurons and/or glial cells and then to be taken up by recipient cells, where the prion-like templating process is repeated. Indeed, numerous biochemical experiments support the idea that TDP-43 protein misfolding can template in trans, leading to propagation of pathological forms of TDP-43 in vitro, and there are elegant experiments that demonstrate inter-cellular spread, both in cell culture and

[1]Department of Anesthesiology, Stony Brook School of Medicine, Stony Brook, NY 11794, USA. [2]Department of Neurobiology and Behavior, Stony Brook University, Stony Brook, NY 11794, USA. ✉e-mail: Yung-Heng.Chang@stonybrook.edu; Joshua.Dubnau@stonybrookmedicine.edu

by injection into cortex of mice that also over-express a sub patholo- gical level of transgenic human TDP-43 variants[3,9–11]. These experi- ments reveal that pathological cytoplasmic hyperphosphorylated TDP-43 appears in cells far from the site of injection. But unlike the case of PrP, injections of pathological TDP-43 does not cause robust propagation of proteinopathy in recipient animals, and does not cause neurodegenerative cellular effects, unless those animals also over- express transgenic human TDP-43 in addition to the endogenous mouse gene[9,11]. We and others have documented that TDP-43 protein pathology or knockout of TDP-43 are each sufficient to cause dys- functional expression of ERVs and retrotransposable elements (RTEs). This has been shown in *Drosophila*, human neuroblastoma cells, and cortical tissue from human subjects[12–21]. Here, we provide evidence that ERVs provide a hitherto unknown component of a positive feed- back loop to sustain propagation of TDP-43 pathology and to spread neurodegenerative effects through neuronal tissue.

## Results

### Human HERV-K and *Drosophila* gypsy-ERV (mdg4-ERV) can trigger TDP-43 pathology in human neuroblastoma and fly S2 cells

In addition to its role in silencing the expression of ERVs and RTEs, TDP-43 also may have a fundamental role in the cellular response to exogenous viral infection[22–24]. Indeed, enteroviral infection can cause pathological accumulation of TDP-43[23,24]. We therefore tested whether expression of ERVs could impact the localization and phosphorylation of TDP-43. We first tested the impact of expressing HERV-K, a human ERV whose expression has been observed in cortical tissue of patients that exhibit TDP-43 pathology[13,15,16]. We generated a HERV-K reporter construct that was tagged with a nuclear localized H2B-mCherry (HERV-K-H2B- mCherry) (Fig. 1a). To accomplish this, we fused a porcine teschovirus-1 2 A (P2A) self-cleaving peptide between HERV-K ORF3 (Env) gene and the H2B-mCherry, thus permitting the expression of the nuclear mCherry without interfering with the production and localization of Env as a separate protein. In HERV- K-H2B-mCherry transfected cells, both Env protein encoded by HERV-K OFR3 and H2B-mCherry were detected in the same cells, but with different localization. The H2B-mCherry signal was localized to the nucleus as expected, but the Env protein (detected with an antibody against HERV-K Env) was found in puncta that appeared to be at or near the cellular membrane (Fig. 1a). In contrast, Env signal was not detected in the control cells that were transfected with the H2B-mCherry (Fig. 1a).

Next, we tested whether HERV-K expression could impact accu- mulation of pathological forms of pTDP-43, we transfected SH-SY5Y human neuroblastoma cells with either the HERV-K-H2B-mCherry or a H2B-mCherry as a control (Fig. 1b). Using an antibody that detects pathological phosphorylation at residues S409/410, we detected a clear increase in cytoplasmically localized pTDP-43 in cells that also exhibit expression of H2B-mCherry associated with the HERV-K reporter (Fig. 1b, c). In contrast, we detected very little pTDP-43 in cells transfected with H2B-mCherry control (Fig. 1b, c). This HERV-K driven accumulation of pathological pTDP-43 also is associated with reduced expression of Stathmin-2 (STMN2) and POLDIP3α (Supple- mentary Fig. 1a–c), two established targets of TDP-43[25,26]. We also observe cytoplasmic mislocalization of TDP43 using a pan-TDP-43 antibody (Supplementary Fig. 1d). Together, these results indicate that expression of HERV-K is sufficient to promote the pathological phos- phorylation of TDP-43 and its redistribution from the nucleus to the cytoplasm, leading to loss of normal TDP-43 function. Similarly, expression of the *Drosophila* ERV gypsy (mdg4) is sufficient to trigger cytoplasmic accumulation of pathologically phosphorylated human TDP-43 (Supplementary Fig. 2a–d) in *Drosophila* S2 cells (Note: The term gypsy is recognized as a racial slur. From this point, we use the

alternate historical name for this ERV, mdg4). Indeed, by 24 h after induced expression of a V5-tagged human TDP-43, 64.31% of cells that were transfected with an mCherry tagged mdg4-ERV exhibited redis- tribution of TDP-43-V5 to the cytoplasm compared with just 35% of cells transfected with a control mCherry (Supplementary Fig. 2d). Thus, as with HERV-K, expression of the mdg4-ERV is sufficient to promote the redistribution of TDP-43 out of the nucleus into cyto- plasmic puncta. By fractionating S2 lysates into RIPA-soluble or urea- soluble fractions[27], we also found that expression of mdg4-ERV also promoted formation of insoluble aggregates of pTDP-43 (Supple- mentary Fig. 2e, f).

The findings indicate a conserved ability of ERV expression to trigger pathological mis-localization, phosphorylation, and aggre- gation of human TDP-43. Because previous work has established that TDP-43 proteinopathy is sufficient to drive expression of ERVs[12–14,18], this suggests the possibility of a reinforcing feedback. We investigated this idea in a *Drosophila* model that has been the source of many of the initial findings regarding the role of RTEs and ERVs in TDP-43 mediated neurodegeneration. Importantly, the fly model also is uniquely suited to manipulate both TDP-43 and ERV expression with cell type specifi- city and temporal control.

### Intercellular propagation of human TDP-43 proteinopathy in a *Drosophila* in vivo model

To mimic the focal onset and intercellular propagation that is hypo- thesized to take place in human patients, we established a *Drosophila* platform to induce human TDP-43 pathology in subsets of glia, and monitor the spread of pathological effects to neurons. Throughout the manuscript, all in vivo *Drosophila* experiments were performed using male flies. We took advantage of a *Drosophila* strain in which the fly TDP-43 homolog (TBPH) has been replaced by a human TDP-43 cDNA with a 3xFlag tag fused at the N-terminal (abbreviated as TDP-43[WT-KI])[28]. In this strain, only the human TDP-43 gene is expressed, at physiolo- gical levels via the endogenous fly ortholog's promoter[28]. Although knockout of the endogenous fly ortholog is lethal, this humanized TDP-43[WT-KI] fly strain exhibits normal longevity that is indistinguishable from that of the laboratory wild-type strain[28] (Supplementary Fig. 3a). Importantly, the TDP-43[WT-KI] fly strain also exhibits normal nuclear localization of hTDP-43, and exhibits no evidence of pathological changes in TDP-43 expression, phosphorylation or localization, even with age[28] (Fig. 2c, d). Within this humanized TDP-43[WT-KI] context, we focally expressed higher levels of a separate UAS-human TDP-43 (hTDP-43) only in a spatially restricted glial subtype called the sub- perineurial glia (SPG) (Fig. 2a, b). It is well established, both in mam- malian and *Drosophila* models, that over-expression of hTDP-43 can trigger nuclear clearance and accumulation of hyperphosphorylated aggregated forms of TDP-43 in the cytoplasm[1]. Such a transgenic overexpression-based approach utilizes non-physiological levels of TDP-43 to trigger pathology, thereby mimicking both the loss of nuclear localization and the accumulation of cytoplasmic aggregates that are seen in human tissues[29,30]. Importantly, we restricted the over- expression to a small subset of glia in order to trigger a focal onset (Fig. 2b). This permitted observation of effects on surrounding neu- rons and glial cells that express only physiological levels of hTDP-43. We chose to focus the over-expression in the SPG in part because we have previously shown that TDP-43 over-expression in this glial cell type induces mdg4-ERV expression and kills nearby neurons[12]. In addition, the SPG as a population consist of only ~300 glia in total out of about ~15,000 glia and ~100,000 neurons in the central brain[31,32]. Thus SPG are sparse enough that they provide the ability to induce isolated focal sources of pathological TDP-43. Finally, the SPG location and morphology is also ideal for detection of non-cell autonomous effects on neurons. In addition to being low in number, the SPG exhibit an elongated planar morphology along the surface of the brain, with many neuronal cell bodies lying just below[12]. This provides the ability

**a**

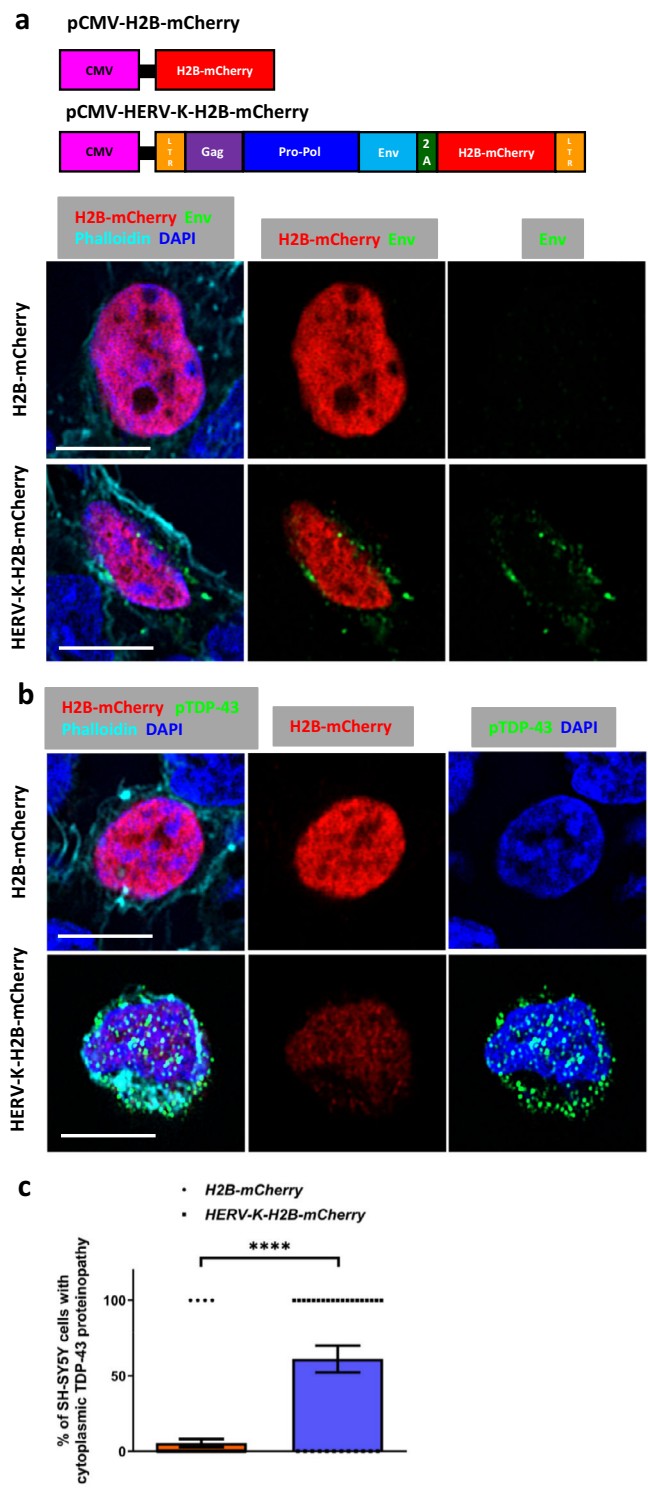

**Fig. 1 | Human HERV-K triggers TDP-43 proteinopathy in human neuro-blastoma cells. a** A HERV-K-H2B-mCherry construct provides a fluorescent reporter of HERV-K expression. H2B-mCherry was used as a control. Immuno-fluorescent images of SH-SY5Y cells transfected with H2B-mCherry or HERV-K-H2B-mCherry and stained for HERV-K Env (green), H2B-mCherry (red), Phalloidin (cyan) and DAPI (blue). Scale bar = 5 μm. mCherry is detected in the nucleus. Env appears as punctate signal at or near the cell membrane. **b** Immunofluorescent images of SH-SY5Y transfected with H2B-mCherry or HERV-K-H2B-mCherry and stained for pTDP-43 (green), H2B-mCherry (red), Phalloidin (cyan) and DAPI (blue). Scale bar = 5 μm. **c** Quantification of cytoplasmic pTDP-43 signal in SH-SY5Y cells 2 days after transfection with H2B-mCherry or HERV-K-H2B-mCherry (5.48 ± 2.68%, $n = 73$; 61.29 ± 8.89%, $n = 31$). Data are shown as mean ± SEM. A two-tailed unpaired t-test was performed. ****$p < 0.0001$. $P$-values for significant comparisons are provided in the Source Data file.

survival of animals that contain the TDP-43[WT-KI] replacement of the fly ortholog. Median survival of this strain was just 18 days compared 31 days for animals containing just the TDP-43[WT-KI] and 34 days for animals that contain the SPG[ts] and TDP-43[WT-KI], but no UAS-hTDP-43 (Supplementary Fig. 3a).

We examined the effects of this focal induction of TDP-43 on the localization and phosphorylation of TDP-43 both within the SPG themselves and in nearby neurons. We used an established antibody that detects pathological human pTDP-43 and found that induction of ectopic hTDP-43 in SPG was sufficient to cause accumulation of pTDP-43 in both the nucleus and cytoplasm of the SPG itself at day 7 post induction (D7) (Fig. 2g–g", Supplementary Fig. 4a, b). This effect appeared to be exacerbated over time (D7 to D15) after induction (Fig. 2g, h', Supplementary Fig. 4a, b). By contrast, we did not detect any evidence of pTDP-43 within SPG of control groups during the same experimental time window (Fig. 2c–f). Next, we examined the localization and phosphorylation status of TDP-43 in surrounding neurons. In contrast with the SPG, in which hTDP-43 is over-expressed to trigger initiation of pathology, the only source of TDP-43 expression in neurons is the human TDP-43[WT-KI], which is expressed at physiological levels under control of the fly TDP-43 gene's promoter. In control animals that did not over-express hTDP-43 in SPG (TDP-43[WT-KI] and SPG[ts] in TDP-43[WT-KI] groups), we observed only nuclear Flag-labeled TDP-43 in neurons, with no evidence of pTDP-43 accumulation. This was true over a 15 day time course after shifting to induction temperature (Fig. 2c–f, j). By contrast, when we induced ectopic TDP-43 in SPG, we detected both nuclear and cytoplasmic pTDP-43 signal in the neurons at D7 (Fig. 2g–g", Supplementary Fig. 4a). This effect was even more dramatic at D15, when we observed pTDP-43 signal in a higher fraction (impact score in Supplementary Fig. 3b, and see methods) of neurons and at an increasing distance (spreading score in Supplementary Fig. 3b, and see methods) from the source of ectopic TDP-43 in the SPG (Fig. 2h, h'; Fig. 2i–k; Supplementary Fig. 4a, b). For example, the spreading score after induction of TDP-43 in SPG increased from 0.88 to 1.61 cell diameters between D7 and D15 (Fig. 2j), and the impact score increased from 11.93% to 25.32% at D7 and D15 (Fig. 2k). We also validated these phenomena using a second pTDP-43 antibody made by a different manufacturer to rule out potential artifacts caused by a single antibody (Supplementary Fig. 5a–e). We also noted that in neurons where we detected pTDP-43, there was a depletion of the nuclear signal derived from the Flag tag (Fig. 2g', g", h'). This indicated that appearance of pTDP-43 also was associated with loss of nuclear localization of the 3xFlag-hTDP-43 knock in allele. Both TDP-43 nuclear clearance and appearance of pTDP-43 are hallmarks of pathological TDP-43 that are seen in ALS and FTD human subjects[27,29,30]. Taken together, these data indicate that induction of ectopic TDP-43 in SPG is sufficient to trigger non-cell-autonomous accumulation of pathological changes in TDP-43 localization and phosphorylation. Over time, such changes occur in increasing fractions of neurons and at increasing distance from the source of ectopic TDP-43.

to observe impacts in neurons at varying distance from the source of pathological TDP-43.

We first tested the impacts of such ectopic expression in SPG on lifespan of the animals. We used a Gal4 line that expresses specifically in SPG together with a temperature sensitive Gal80 (SPG[ts]), so that we could induce spatially restricted TDP-43 expression after allowing normal development (induction relies on shifting the growth temperature from 21 °C to 29 °C). The use of an inducible expression system also provides the means for observing effects over a time-course after initiating expression. We found that Gal4 mediated induction of hTDP-43 in SPG (SPG[ts] > hTDP-43) greatly reduces median

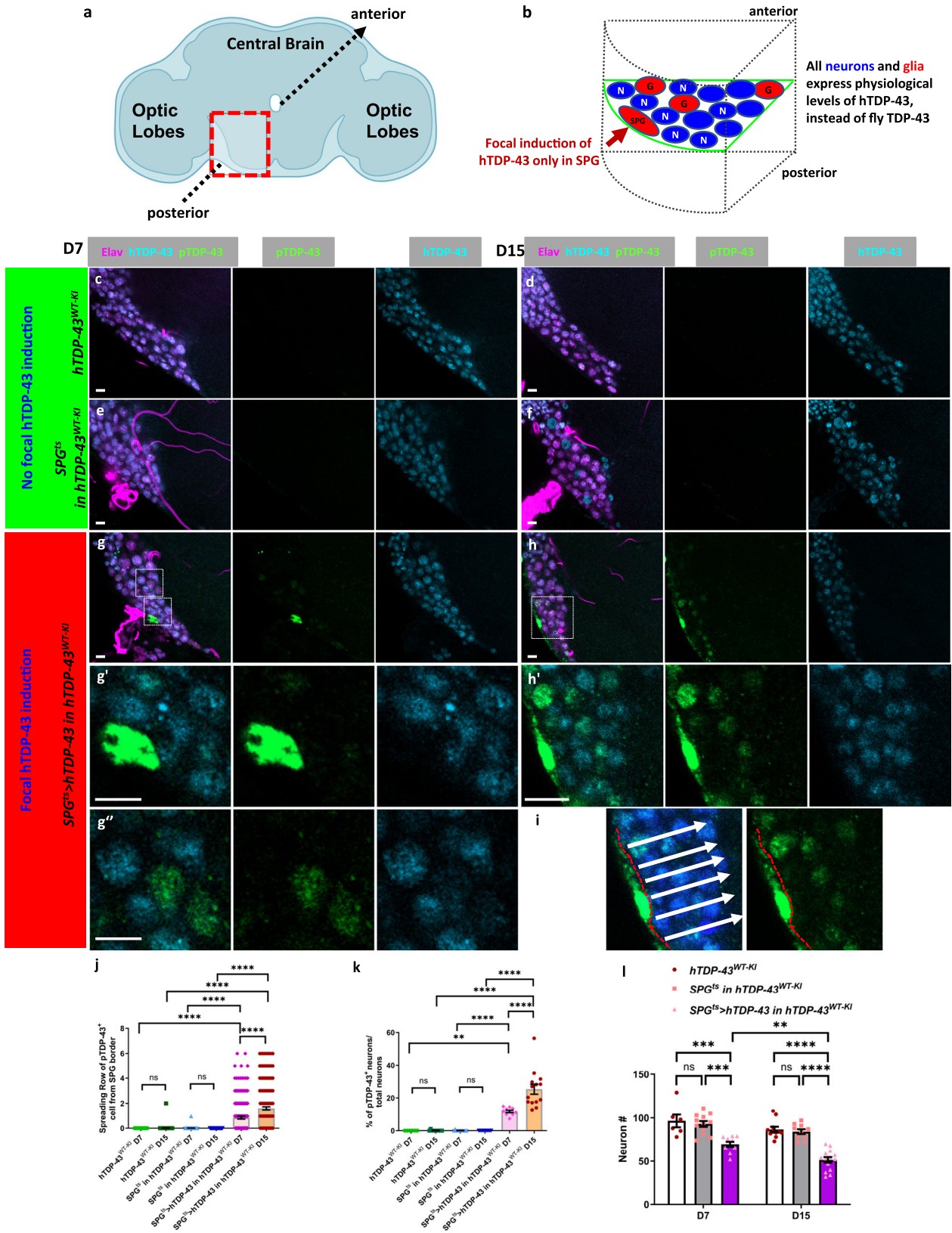

## TDP-43 pathology in SPG causes neuronal DNA damage and neuronal loss

Loss of nuclear TDP-43 has been found to cause DNA damage due to either a defect in repair of DNA breaks resulting from R-loop formation, or by impairing the non-homologous end joining repair machinery[33–36]. DNA damage has been observed in post mortem brain tissue from patients with TDP-43 related disorders[36], in iPSC derived neurons from C9orf72 familial patients that exhibit TDP-43 pathology[37], and in animal models including *Drosophila* when TDP-43 is over-expressed[12,34]. We therefore wondered whether we would detect DNA damage foci in neurons that exhibited pTDP-43 pathology due to focal induction of hTDP-43 over-expression in SPG. Importantly,

**Fig. 2 | Induction of TDP-43 proteinopathy in *Drosophila* surface glia causes spread of pTDP-43 to neurons that express physiological levels of human TDP-43. a** Schematic of *Drosophila* brain showing region where all images were taken (made in ©BioRender–biorender.com). **b** schematic depicting the relative location of SPG and other cell types as depicted in fluorescent images. Neurons (N), and other glial subtypes (G). **c–h** Immunofluorescent images from a central brain section in the region indicated in b, with staining for pTDP-43 (green), human TDP-43$^{WT-KI}$ Flag tag (cyan) and neuronal cell marker Elav (violet). Scale bar = 5 µm. **c, d** TDP-43$^{WT-KI}$ at D7 and D15. **e, f** SPG$^{ts}$ in human TDP-43$^{WT-KI}$ background at D7 and D15. **g, h** SPG$^{ts}$ > hTDP-43 in human TDP-43$^{WT-KI}$ background at D7 and D15. **g'–g"**, Enlarged images from selected regions indicated in **g. h'** Enlarged images from the selected region indicated in **h. i** Schematic description of spreading score counting. **j** Quantification of pTDP-43 spreading scores of human TDP-43$^{WT-KI}$, SPG$^{ts}$ in human TDP-43$^{WT-KI}$ background and SPG$^{ts}$ > hTDP-43 in human TDP-43$^{WT-KI}$ background at D7 (0.00 ± 0.00, $n = 86$; 0.01 ± 0.01, $n = 171$; 0.88 ± 0.11, $n = 183$) and D15 (0.01 ± 0.01, $n = 170$; 0.00 ± 0.00, $n = 144$; 1.61 ± 0.13, $n = 196$). Data are shown as mean ± SEM and two-way ANOVA with Tukey's multiple comparisons was performed. ****$p < 0.0001$; ns no significant difference. **k** Quantification of pTDP-43 impact scores of human TDP-43$^{WT-KI}$, SPG$^{ts}$ in human TDP-43$^{WT-KI}$ background and SPG$^{ts}$ > hTDP-43 in human TDP-43$^{WT-KI}$ background at D7 (0.00 ± 0.00%, $n = 6$; 0.08 ± 0.08%, $n = 12$; 11.93 ± 0.71%, $n = 10$) and D15 (0.10 ± 0.10%, $n = 11$; 0.00 ± 0.00%, $n = 10$; 25.32 ± 3.03%, $n = 14$). Data are shown as mean ± SEM and a two-way ANOVA with Tukey's multiple comparisons was performed. **$p < 0.01$; ****$p < 0.0001$; ns, no significant difference. **l** Quantification of neuronal cell numbers at selected sections of human TDP-43$^{WT-KI}$, SPG$^{ts}$ in human TDP-43$^{WT-KI}$ background and SPG$^{ts}$ > hTDP-43 in human TDP-43$^{WT-KI}$ backgrounds at D7 (96.50 ± 7.36, $n = 6$; 93.00 ± 3.32, $n = 12$; 69.70 ± 2.90, $n = 10$) and D15 (86.73 ± 3.14, $n = 11$; 84.20 ± 2.71, $n = 10$; 51.36 ± 3.21, $n = 14$). Data are shown as mean ± SEM and a two-way ANOVA with Tukey's multiple comparisons was performed. **$p < 0.01$; ***$p < 0.001$; ****$p < 0.0001$; ns, no significant difference. *P*-values for significant comparisons are provided in the Source Data file.

such neurons exhibit pTDP-43 due to induced hTDP-43 over-expression in a neighboring cell type while only expressing human TDP-43 at physiological levels themselves (Fig. 2). To monitor DNA damage, we used an established antibody against γH2Av[38], a *Drosophila* ortholog of the mammalian DNA damage marker, γH2AX. In control animals that only contained the human TDP-43$^{WT-KI}$, γH2Av signal was rarely detected over a 15-day experimental course (Supplementary Fig. 5a, b'). By contrast, we detected a significant increase in the number of γH2Av$^+$ neurons at D7 in the SPG$^{ts}$ > TDP-43 in TDP-43$^{WT-KI}$ background group. The fraction of γH2Av$^+$ neurons in these animals increased dramatically by D15 (Supplementary Fig. 6c, d). Notably, neurons with increased γH2Av$^+$ signal also exhibited a depletion of Flag signal (Supplementary Fig. 6c', d'), indicating a loss of nuclear localization of the flag-tagged TDP-43$^{WT-KI}$. This effect was similar to what we observed in neurons that exhibit pTDP-43 signal (Fig. 2g', h'). We used the same spreading score and impact score to quantify the average distance from the SPG and the fraction of neurons exhibiting DNA damage signal. In the SPG$^{ts}$ > TDP-43 in TDP-43$^{WT-KI}$ background group, the spreading scores were 1.31 and 3.11 cell diameters at D7 and D15 (Supplementary Fig. 6e), indicating DNA damage occurred in the neurons at some distance from the initiating source of TDP-43 proteinopathy from induction in SPG. By contrast, spreading scores in the control group at D7 and D15 were 0.04 and 0.03 cell diameters (Supplementary Fig. 6e), indicating no evidence of DNA damage being triggered. Impact scores (fraction of neurons exhibiting DNA damage) for TDP-43$^{WT-KI}$ control vs SPG$^{ts}$ > TDP-43 in TDP-43$^{WT-KI}$ were 0.42% vs 31.56% at D7 and 0.42% vs 83.54% at D15 (Supplementary Fig. 5f). We also quantified the effects of inducing TDP-43 in SPG on neuronal survival. We found that induction of TDP-43 in the SPG$^{ts}$ > TDP-43 in TDP-43$^{WT-KI}$ background group causes a gradual loss of neurons over time (Fig. 2l).

## No detectable intercellular movement of TDP-43 from SPG to neurons

One possible explanation for the appearance of pTDP-43 pathology in neurons is the intercellular transmission of seeds of pathological TDP-43 from the SPG to the neurons, which then might trigger re-initiation of TDP-43 aggregation[3]. To examine this possibility, we induced TDP-43 in SPG, but in animals that contained the endogenous fly TDP-43 ortholog (TBPH) rather than the humanized TDP-43$^{WT-KI}$ knock in (Fig. 3a). This allowed us to faithfully detect pathological pTDP-43 that might be secreted from the SPG source without background signal of pan neuronal human TDP-43 in the TDP-43$^{WT-KI}$ strain. In order to maximize our chances of detecting any movement of the induced human-TDP-43 from SPG, we induced an RFP-tagged hTDP-43 (hTDP-43-RFP)[39] in SPG using the same Gal80$^{ts}$ approach. This allowed us to monitor both pTDP-43 signal and the RFP signal independently. We also co-expressed a membrane tethered mCD8-GFP in the SPG in order

to mark the membrane, yielding an outline of SPG morphology. As with induction of the untagged TDP-43, induction in SPG of the TDP-43-RFP (SPG$^{ts}$ > hTDP-43-RFP in TDP-43$^{WT-KI}$) shortened the animal's lifespan to a median survival of 15 days (Supplementary Fig. 3a).

Given the similar effects on lifespan from expressing hTDP-43 and the RFP-tagged hTDP-43, we examined the effects on TDP-43 pathology. Induction of ectopic hTDP-43-RFP in SPG (SPG$^{ts}$ > hTDP-43-RFP + mCD8-GFP) resulted in a strong cell autonomous effect on TDP-43 pathology, resulting in robust pTDP-43 signal in all SPG at D7, and this persisted to D15 (Fig. 3d–f, Supplementary Fig. 6a). By contrast, we do not detect any evidence of pTDP-43 in SPG with expression of the myr-RFP control (SPG$^{ts}$ > myr-RFP + mCD8-GFP) (Fig. 3b, c, f, Supplementary Fig. 7a). Such ectopic hTDP-43-RFP in SPG of animals that expressed only the fly ortholog in neurons also shortened lifespan similar to the effects of expression in animals that had the human TDP-43$^{WT-KI}$ (SPG$^{ts}$ > hTDP-43-RFP + mCD8-GFP with 19.5 days and SPG$^{ts}$ > myr-RFP + mCD8-GFP with 29 days of median survival) (Supplementary Fig. 7b). Thus, induction in SPG of the RFP-tagged human TDP-43 caused similar effects on animal survival and pTDP-43 accumulation in the SPG, irrespective of whether the neurons expressed the native fly TDP-43 ortholog or the human gene knocked into the endogenous locus.

Next, we turned to examine non-cell autonomous effects on neurons from TDP-43-RFP induction in SPG. We were unable to detect pTDP-43 or RFP signal in the surrounding neurons during a 15 day experimental time course (Fig. 3d-e'). This was true even in neurons that were immediately adjacent to the SPG that exhibited ectopic hTDP-43-RFP and strong pTDP-43 pathology (Fig. 3d', e'). The spreading scores for pTDP-43 were 0 cell diameters at both D7 and D15 after induction of either ectopic hTDP-43-RFP, or the myr-RFP control (Fig. 3g). We also measured the spreading scores of RFP-labeled proteins (myr-RFP and hTDP-43-RFP) and observed no significant spread to neurons of either of these proteins during the experimental course (Fig. 3h). Since there was no detectable pTDP-43 or RFP signal observed in the neurons, pTDP-43 and RFP-labeled proteins' impact scores for both myr-RFP and hTDP-43-RFP groups were 0% at each time point (Fig. 3i, j). This is in stark contrast to the strong pTDP-43 signal that we detect in neurons at a distance from the SPG when the neurons express physiological levels of the human TDP-43$^{WT-KI}$ instead of the fly gene (Fig. 2).

Although we were unbale to detect any spread of hTDP-43-RFP protein or human pTDP-43 signal from SPG, where TDP-43 pathology was induced, to neurons that express only the fly ortholog, it was notable that these animals exhibited an equivalent reduction of lifespan as animals that had the human TDP-43$^{WT-KI}$ (Supplementary Fig. 2a, Supplementary Fig. 7b). We therefore examined whether DNA damage and neuronal loss occurred in the surrounding neurons as we had previously observed in the human TDP-43$^{WT-KI}$ context. Indeed

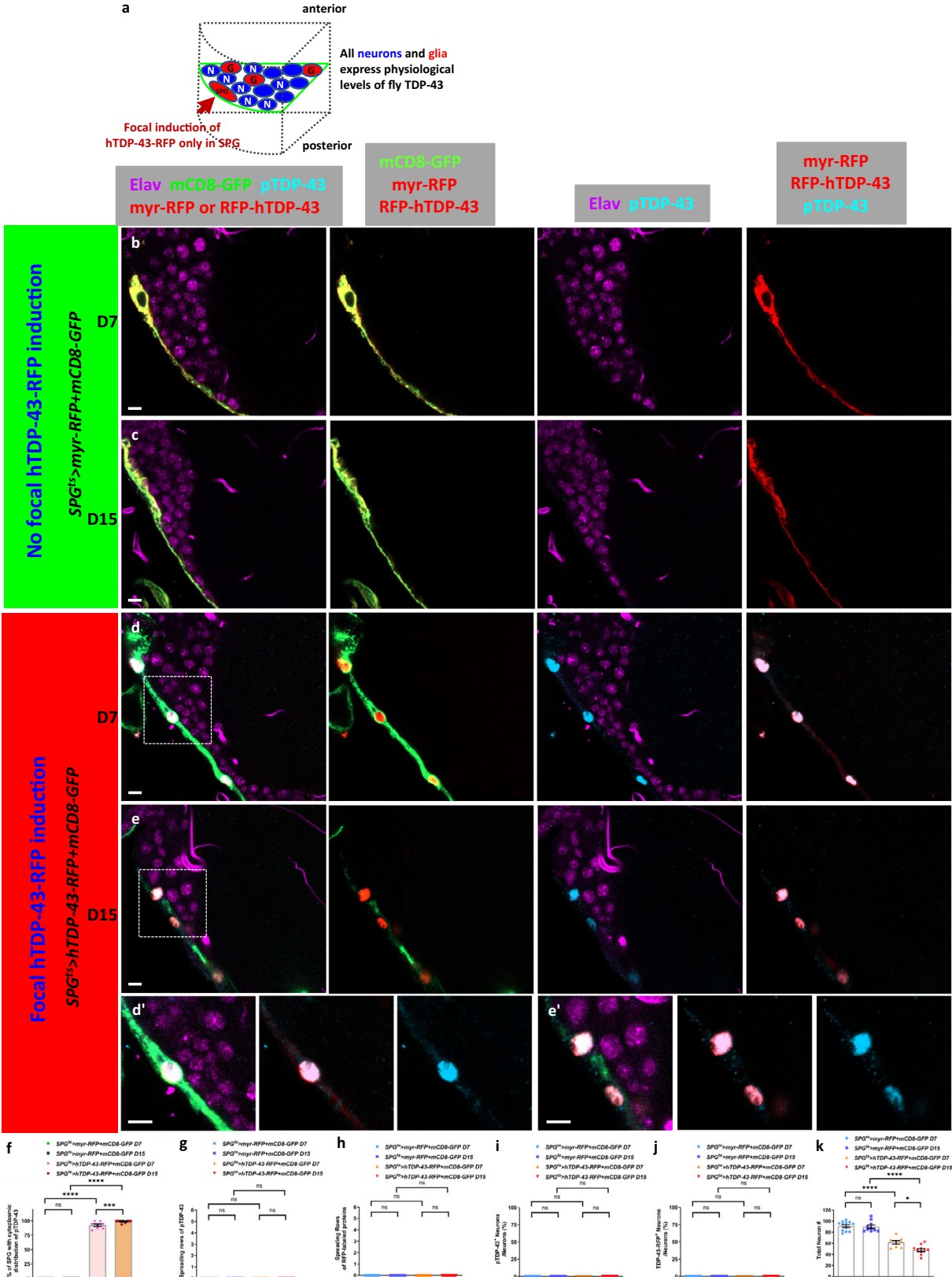

we detected γH2Av+ neurons at both D7 and D15 after induction in experimental SPGts > hTDP-43-RFP + mCD8-GFP brains (Supplementary Fig. 7c, d'). As expected, γH2Av+ neurons were rarely found in control SPGts > myr-RFP + mCD8-GFP group (Supplementary Fig. 8a, b). In the myr-RFP control, the spreading score at D7 and D15 were 0.04 and 0.03 cell diameters, but they were increased to 1.67 (D7)

and 3.06 (D15) cell diameters in the animals where hTDP-43-RFP was induced in SPG (Supplementary Fig. 8e). The impact scores also exhibited a similar effect during the experimental time course (0.33% and 0.62% in control myr-RFP and 45.88% and 84.22% in experimental hTDP-43-RFP) (Supplementary Fig. 7f). And we also observed a significant loss of neurons with ectopic hTDP-43-RFP in SPG (Fig. 3k).

**Fig. 3 | Human TDP-43 proteinopathy in surface glia is toxic to neurons at a distance, but without detectable spread of TDP-43. a** Schematic depicting the relative location of SPG and other cell types as depicted in fluorescent images. **b–e** Immunofluorescence images from the central brain with staining for myr-RFP or RFP-hTDP-43 (red), pTDP-43 (cyan), mCD8-GFP (green) and neuronal cell marker Elav (violet). Scale bar = 5 μm. **b, c** SPG$^{ts}$ > myr-RFP + mCD8-GFP at D7 (**b**) and D15 (**c**). **d, e** SPG$^{ts}$ > hTDP-43-RFP + mCD8-GFP at D7 (**d**) and D15 (**c**). **d', e'** Enlarged images from indicated regions at **d** and **e**. Scale bar = 5 μm. **f** Quantification of percentage of SPG that exhibit TDP-43 proteinopathy after ectopic myr-RFP or hTDP-43-RFP in SPG$^{ts}$ > myr-RFP + mCD8-GFP and SPG$^{ts}$ > hTDP-43-RFP + mCD8-GFP groups at D7 (0.00 ± 0.00%, n = 12; 92.51 ± 2.20%, n = 8) and D15 (0.00 ± 0.00%, n = 10; 99.05 ± 0.67%, n = 10). Data are shown as mean ± SEM and two-way ANOVA with Tukey's multiple comparisons was performed. ***p < 0.001; ****p < 0.0001; ns, no significant difference. **g** Quantification of pTDP-43 spreading scores of SPG$^{ts}$ > myr-RFP + mCD8-GFP and SPG$^{ts}$ > hTDP-43-RFP + mCD8-GFP groups at D7 (0.00 ± 0.00, n = 171; 0.00 ± 0.00, n = 115) and D15 (0.00 ± 0.00, n = 142; 0.00 ± 0.00, n = 93). Data are shown as mean ± SEM and a two-tailed Wilcoxon signed-rank test was performed. ns, no significant difference. **h** Quantification of RFP-labeled protein

spreading scores in SPG$^{ts}$ > myr-RFP + mCD8-GFP and SPG$^{ts}$ > hTDP-43-RFP + mCD8-GFP at D7 (0.00 ± 0.00, n = 171; 0.00 ± 0.00, n = 115) and D15 (0.00 ± 0.00, n = 142; 0.00 ± 0.00, n = 93). Data are shown as mean ± SEM and a two-tailed Wilcoxon signed-rank test was performed. ns, no significant difference. **i** Quantification of pTDP-43 impact scores of SPG$^{ts}$ > myr-RFP + mCD8-GFP and SPG$^{ts}$ > hTDP-43-RFP + mCD8-GFP at D7 (0.00 ± 0.00%, n = 12; 0.00 ± 0.00%, n = 8) and D15 (0.00 ± 0.00%, n = 10; 0.00 ± 0.00%, n = 10). Data are shown as mean ± SEM and a two-tailed Wilcoxon signed-rank test was performed. ns, no significant difference. **j** Quantification of RFP-labeled protein impact scores of SPG$^{ts}$ > myr-RFP + mCD8-GFP and SPG$^{ts}$ > hTDP-43-RFP + mCD8-GFP at D7 (0.00 ± 0.00%, n = 12; 0.00 ± 0.00%, n = 8) and D15 (0.00 ± 0.00%, n = 10; 0.00 ± 0.00%, n = 10). Data are shown as mean ± SEM and a two-tailed Wilcoxon signed-rank test was performed. ns, no significant difference. **k** Quantification of neuronal cell number at selected section of SPG$^{ts}$ > myr-RFP + mCD8-GFP and SPG$^{ts}$ > hTDP-43-RFP + mCD8-GFP at D7 (91.00 ± 2.40, n = 12; 61.63 ± 3.26, n = 8) and D15 (89.90 ± 3.01, n = 10; 47.60 ± 2.76, n = 10). Data are shown as mean ± SEM and two-way ANOVA with Tukey's multiple comparisons was performed. *p < 0.05; ****p < 0.0001; ns, no significant difference. *P*-values for significant comparisons are provided in the Source Data file.

These findings highlight the fact that the non-cell autonomous effects on neurons from induced human TDP-43 in SPG are equivalent regardless of whether the neurons express physiological levels of the human TDP-43$^{WT-KI}$ or the native fly ortholog. The pTDP-43 antibodies that are available do not detect pathological effects on the fly ortholog. Thus, we cannot address the possibility that the endogenous fly ortholog is caused to accumulate in cytoplasmic aggregates in neurons. If true, this could be caused either by transfer of misfolded human TDP-43 that seeds aggregation of the fly gene, or by release of a different factor. Because we were unable to detect transfer of hTDP-43-RFP or pTDP-43 from SPG to neurons, we considered the second possibility.

### mdg4-ERV expression in SPG is required to trigger neuronal pTDP-43, neuronal DNA damage and neuronal loss

We previously demonstrated that focal induction of hTDP-43 in small subsets of glia is sufficient to cause death of nearby neurons and that expression and/or replication of mdg4-ERV causally contributes to these effects[12]. We also reported that in cell culture[40], mdg4-ERV is capable of viral transmission, as is the case for HERV-K[16,41]. Given the observation that either HERV-K or mdg4 expression are sufficient to trigger pTDP-43 in cell culture (Fig. 1, Supplementary Fig. 2), we therefore wondered whether mdg4-ERV activation in SPG might play a role in triggering the appearance of pTDP-43 in neurons. We used a previously established double-strand RNA (dsRNA) transgene (UAS-mdg4-IR) to that is capable of knocking down levels of mdg4 in SPG[12,14,42,43] (Fig. 4a). As a control, we used a dsRNA targeting GFP (UAS-GFP-IR)[12,14,42] (Fig. 4a). In the SPG$^{ts}$ > hTDP-43+GFP-IR in TDP-43$^{WT-KI}$ animals, that expressed a control RNAi, we detected robust pTDP-43 signal in neighboring neurons with signal detected at increasing distance (1.86 vs 0.89 cell diameters) and in a higher fraction of cells at (29.02% vs 12.12%) at D15 than at D7 (Fig. 4b, c, f, g). These effects are similar to those described above (Fig. 2). By contrast, when we knocked down the expression of mdg4-ERV within the TDP-43 expressing SPG animals (SPG$^{ts}$ > hTDP-43 + mdg4-IR in human TDP-43$^{WT-KI}$), we significantly attenuated these non-cell autonomous effects on pTDP-43 in neurons (Fig. 4d–g). With mdg4-ERV knockdown, the spreading scores for pTDP-43 signal at D7 and D15 were 0.35 and 0.69 cell diameters, markedly lower than that seen with the SPG$^{ts}$ > hTDP-43+GFP-IR in human TDP-43$^{WT-KI}$ background group (Fig. 4f). There also was a significant decrease in the fraction of neurons and glia that exhibited pTDP-43 at these time points (Fig. 4g, i). These findings indicate that knocking down the expression of mdg4-ERV within the SPG glia where hTDP-43 pathology is initiated, is sufficient to ameliorate the appearance of TDP-43 pathology in the surrounding neurons that express only physiological levels of human TDP-43$^{WT-KI}$.

We also assessed the effect of mdg4-ERV expression in SPG on the appearance of DNA damage in neurons. As expected, we detected an increase in γH2Av$^+$ neurons in the SPG$^{ts}$ > hTDP-43+GFP-IR in human TDP-43$^{WT-KI}$ animals with spreading scores of 1.37 and 2.41 cell diameters and impact scores of 48.61% and 68.33% at D7 and D15 respectively (Supplementary Fig. 9a, b, e, f). However, when we knocked down mdg4-ERV expression in the SPG via expression of the mdg4-IR (SPG$^{ts}$ > hTDP-43 + mdg4-IR in human TDP-43$^{WT-KI}$ animals), we greatly reduced the appearance of DNA damage in the nearby neurons both at D7 and D15 (Supplementary Fig. 9c–f). Knockdown of mdg4-ERV also significantly reduced the appearance of DNA damage within surrounding glia of other sub-types (Supplementary Fig. 9g). And finally, knockdown of mdg4-ERV expression in SPG also extended lifespans of the animals (Supplementary Fig. 9h), consistent with our prior report that mdg4-ERV contributes causally to the toxicity of glial TDP-43 pathology[12,14].

We next tested whether mdg4-ERV expression also contributed to the non-cell autonomous effects on neuronal loss. In the control SPG$^{ts}$ > hTDP-43+GFP-IR in human TDP-43$^{WT-KI}$ animals, we observed a significant loss of neurons (Fig. 4h). Thus as expected, the presence of the GFP-IR did not prevent the appearance of pTDP-43 in neurons (Fig. 4b, c), and also did not prevent neuronal loss (Fig. 4h). On the other hand, co-expression of the mdg4-IR to knock down the expression of mdg4-ERV (SPG$^{ts}$ > hTDP-43 + mdg4-IR in human TDP-43$^{WT-KI}$ animals) was sufficient to greatly ameliorate the loss of neurons (Fig. 4h). Thus, expression of mdg4-ERV mediates toxic effects of TDP-43 within the SPG and also mediates appearance of pTDP-43 and DNA damage in surrounding neurons and other glial cell types and to neuronal loss.

### Intercellular viral transmission of mdg4-ERV is sufficient to drive propagation of TDP-43 pathology in cell culture

Both the *Drosophila* mdg4-ERV[40,44,45] and human HERV-K[41,46], produce viral-like particles (VLPs) that are capable of transmitting between cells grown in culture. Indeed, both of these ERVs contain an enveloped glycoprotein that is required to mediate such intercellular transmission. We previously established *Drosophila* S2 cell culture assays to detect transmission of mdg4-ERV between cells grown in contact, or between cells that were separated by a 0.4 μm filter in a transwell system[40]. We used these assays to test whether viral transmission of mdg4-ERV was sufficient to mediate propagation of TDP-43 pathology to recipient cells.

As described above, mdg4-ERV expression is sufficient to trigger cytoplasmic translocation and accumulation of insoluble TDP-43 when the mdg4-ERV-mCherry and an inducible hTDP-43-V5 (pMT-hTDP-43-V5) were co-transfected into S2 cells (Supplementary Fig. 2). We used

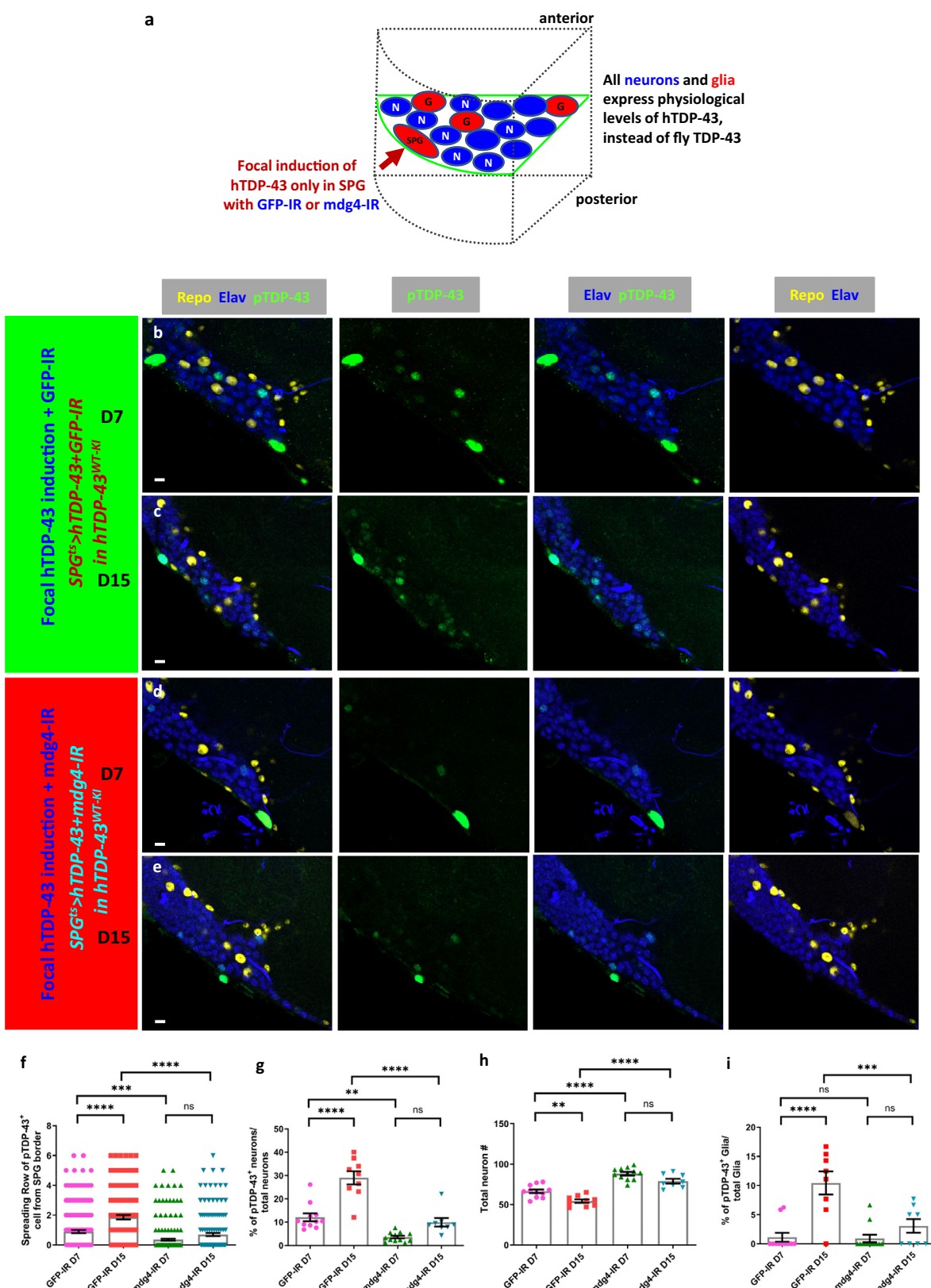

these same constructs to transfect two populations of S2 cells. The first population was transfected with either the mdg4-ERV-mCherry (pAc-mdg4-H2B-mCherry) or the H2B-mCherry (pAc-H2B-mCherry) as a control. The second population of S2 cells was transfected with the inducible hTDP-43-V5 (pMT-hTDP-43-V5). We first tested if mdg4-ERV producing cells were able to cause TDP-43 pathology after

transmitting mCherry tagged virus to recipient cells that were grown in contact (Supplementary Fig. 10a). Indeed, we found that a fraction of the hTDP-43-V5 transfected cells exhibited mCherry signal when they were grown in contact with the mdg4-H2B-mCherry expressing cells (Fig. 5a). This viral transmission is as we previously reported[40]. In contrast, the H2B-mCherry control was never seen to transfer to the

**Fig. 4 | Mdg4-ERV expression in surface glia triggers human TDP-43 proteinopathy in neurons that express only physiological levels of human TDP-43.**
**a** Schematic depicting the relative location of SPG and other cell types as depicted in fluorescent images. **b–e** Immunofluorescent images from central brain sections with staining for pTDP-43 (green), neuronal cell marker Elav (blue) and glial cell marker Repo (yellow). Scale bar = 5 μm. **b, c** SPG$^{ts}$ > hTDP-43+GFP-IR in TDP-43$^{WT-KI}$ background at D7 (**a**) and D15 (**b**). **d, e** SPG$^{ts}$ > hTDP-43 + mdg4-IR in human TDP-43$^{WT-KI}$ background at D7 (**c**) and D15 (**d**). **f** Quantification of pTDP-43 spreading scores of SPG$^{ts}$ > hTDP-43+GFP-IR in human TDP-43$^{WT-KI}$ background and SPG$^{ts}$ > hTDP-43 + mdg4-IR in human TDP-43$^{WT-KI}$ background at D7 (0.89 ± 0.11, n = 183; 0.35 ± 0.07, n = 215) and D15 (1.86 ± 0.14, n = 147; 0.69 ± 0.11, n = 154). Data are shown as mean ± SEM and a two-way ANOVA with Tukey's multiple comparisons was performed. ***p < 0.001; ****p < 0.0001; ns, no significant difference.
**g** Quantification of pTDP-43 impact scores of SPG$^{ts}$ > hTDP-43+GFP-IR in human TDP-43$^{WT-KI}$ background and SPG$^{ts}$ > hTDP-43 + mdg4-IR in human TDP-43$^{WT-KI}$ background at D7 (12.12 ± 1.71%, n = 11; 3.53 ± 0.59%, n = 12) and D15 (29.02 ± 2.84%,

n = 9; 9.92 ± 1.85%, n = 8). Data are shown as mean ± SEM and a two-way ANOVA with Tukey's multiple comparisons was performed. **p < 0.01; ****p < 0.0001; ns, no significant difference. **h** Quantification of neuronal cell number at selected sections of SPG$^{ts}$ > hTDP-43+GFP-IR in human TDP-43$^{WT-KI}$ background and SPG$^{ts}$ > hTDP-43 + mdg4-IR in human TDP-43$^{WT-KI}$ background at D7 (66.36 ± 2.33, n = 11; 88.17 ± 2.07, n = 12) and D15 (54.33 ± 2.15, n = 9; 79.00 ± 2.97, n = 8). Data are shown as mean ± SEM and a two-way ANOVA with Tukey's multiple comparisons was performed. **p < 0.01; ****p < 0.0001; ns, no significant difference. **i** Quantification of pTDP-43 impact scores on glial cells of SPG$^{ts}$ > hTDP-43+GFP-IR in human TDP-43$^{WT-KI}$ background and SPG$^{ts}$ > hTDP-43 + mdg4-IR in human TDP-43$^{WT-KI}$ background at D7 (1.10 ± 0.74%, n = 11; 0.90 ± 0.63%, n = 12) and D15 (10.44 ± 1.98%, n = 8; 3.05 ± 1.19%, n = 8). Data are shown as mean ± SEM and a two-way ANOVA with Tukey's multiple comparisons was performed. ***p < 0.001; ****p < 0.0001; ns, no significant difference. *P*-values for significant comparisons are provided in the Source Data file.

hTDP-43-V5 expressing population of cells that are grown in contact (Fig. 5a). More importantly, we found that hTDP-43-V5 recipient cells that were grown in contact with mdg4-H2B-mCherry cells exhibited a significant increase in cytosolic translocation of TDP-43 and the appearance of pTDP-43 signal (Fig. 5a). At 20 and 24 h after induction of TDP-43-V5, we saw cytosolic TDP-43 puncta and pTDP-43 in 54 and 65% of recipient cells grown in contact with mdg4-H2B-mCherry cells, and just 21 and 40% of cells that were grown in contact with the H2B-mCherry controls (Fig. 5a, c).

Finally, we tested whether transmission of mdg4-ERV could also trigger TDP-43 pathology in recipient cells grown in a transwell system in which the two populations of cells were separated by a membrane with 0.4 μm permeable pores (Supplementary Fig. 10b). Here too, we found that when hTDP-43-V5 cells were grown in opposition to mdg4-ERV-mCherry expressing cells, the percentage of recipient cells that exhibited cytoplasmic TDP-43 was increased from 29.39% (pMT-hTDP-43-V5 + pAc-H2B-mCherry) to 64.31% (pMT-hTDP-43-V5 + pAc-mdg4-H2B-mCherry) after 20 h induction of hTDP-43-V5 and from 37.50% to 67.85% at 24 h (Fig. 5b, d). The hTDP-43-V5 recipient cells in which we were able to detect the mCherry reporter also exhibit robust cytoplasmic pTDP-43 signal (Fig. 5b). This inter-cellular effect on TDP-43 pathology in 'recipient cells' was ameliorated when we used mdg4-ERV producing cells in which we had deleted the mdg4-ORF3, which encodes the viral ENV glycoprotein (Supplementary Fig. 10c, d). Taken together, these results support the hypothesis that viral transmission of the mdg4-ERV triggers the appearance of TDP-43 protein pathology in recipient cells.

## Discussion

Neurodegenerative disorders as diverse as AD, FTD and ALS share some key features[2,4–6]: First, risk of disease onset rises with each passing decade of age. Second, once pathological processes are triggered, clinical progression is relatively rapid. Third, disease pathology is thought to initiate focally but spreads through neural tissue over time. These core characteristics need to be accounted for by any mechanistic model. Upstream triggers likely include phenomena associated with normal aging. Downstream cellular mechanisms are non-cell autonomous but also are robustly sustained as toxicity spreads from cell-to-cell. An example of such a self-sustaining mechanism is the ability of PrP, which underlies infectious prion diseases, to undergo pathological conformational changes, to template this effect in trans, and to move intercellularly[7,8]. This combination of properties is thought to provide a robust, self-sustaining spread through neural tissue. This robustness is evidenced by the fact that either ingestion, or injection of pathological PrP seeds into the brain, are sufficient to cause both propagation of the protein pathology and the full suite of neurodegenerative effects, ultimately killing the animal. By contrast, with the case of the prion-

like TDP-43 protein[2,5,6,9,11], injection into brain of pathological misfolded protein is not sufficient to cause disease.

Many elegant experiments demonstrate that pathologically misfolded TDP-43 isolated from patient tissue is capable of seeding the propagation of protein pathology in culture and by delivery to mouse brain[3,9,10]. But unlike PrP, the ability to propagate cytoplasmic phosphorylated protein to new cells, or to cause neuronal loss or neurological decline requires that the injected animals also over-express pan-neuronal hTDP-43 at sub-pathological levels[9,11]. This suggests that prion-like spread of TDP-43 is insufficient to sustain progression of disease on its own.

We propose that a positive feedback between ERVs and TDP-43 pathology can provide a mechanism for sustained intercellular propagation of neurodegeneration (Fig. 5e). We and others have previously demonstrated that TDP-43 dysfunction causes increased expression of RTEs and ERVs[12–21]. It is notable that HERV-K proteins localize to the same cells as does TDP-43 pathology in patient cortical tissue[13,16]. We also have previously demonstrated in a *Drosophila* model that RTE/ERV expression mediates many of the neurodegenerative effects of TDP-43 dysfunction[12,14]. Here, we demonstrate that human HERV-K or the *Drosophila* mdg4-ERV also are upstream drivers of TDP-43 pathology, and are sufficient to trigger dysfunctional phosphorylation, cytoplasmic translocation, and formation of insoluble forms of TDP-43. Thus, TDP-43 pathology stimulates ERV expression[12–14,16,18] and ERV expression triggers pathological changes in TDP-43. We also demonstrate that inter-cellular viral transmission of mdg4 is sufficient to re-initiate TDP-43 pathology in recipient cells grown in culture. The ERV-TDP-43 feedback mechanism underlies propagation of both TDP-43 pathology and neurodegeneration in a *Drosophila* in vivo context, consistent with the idea that viral transmission also is at play.

Normal aging is associated with a progressive rise in RTE and ERV expression[47–53] and in loss of proteostasis[54,55]. Thus, aging may drive both ERV expression and TDP-43 proteinopathy, leading to an increased risk of triggering a sustaining feedback and disease onset[42,47]. Although our findings focus on HERV-K, it is important to note that human LINE-1 retrotransposons also are expressed in cells that lose nuclear TDP-43 function[15,17]. It will therefore be important to investigate whether LINE-1 expression also can trigger TDP-43 pathology. Finally, a body of literature has established that RTEs and ERVs become inappropriately expressed not only in TDP-43 related disorders ALS and FTD[12–21,56,57], but also in some disorders with Tau pathology (AD and Progressive supranuclear palsy)[58,59] and in Aicardi-Goutieres syndrome[60,61]. RTE and ERV expression has the potential to contribute to many of the cellular impacts that are thought to be at play across these disorders, including inflammatory signaling, DNA damage and defects in RNA metabolism and proteostasis[62–65]. But our findings raise the possibility that RTE and ERV expression provides a

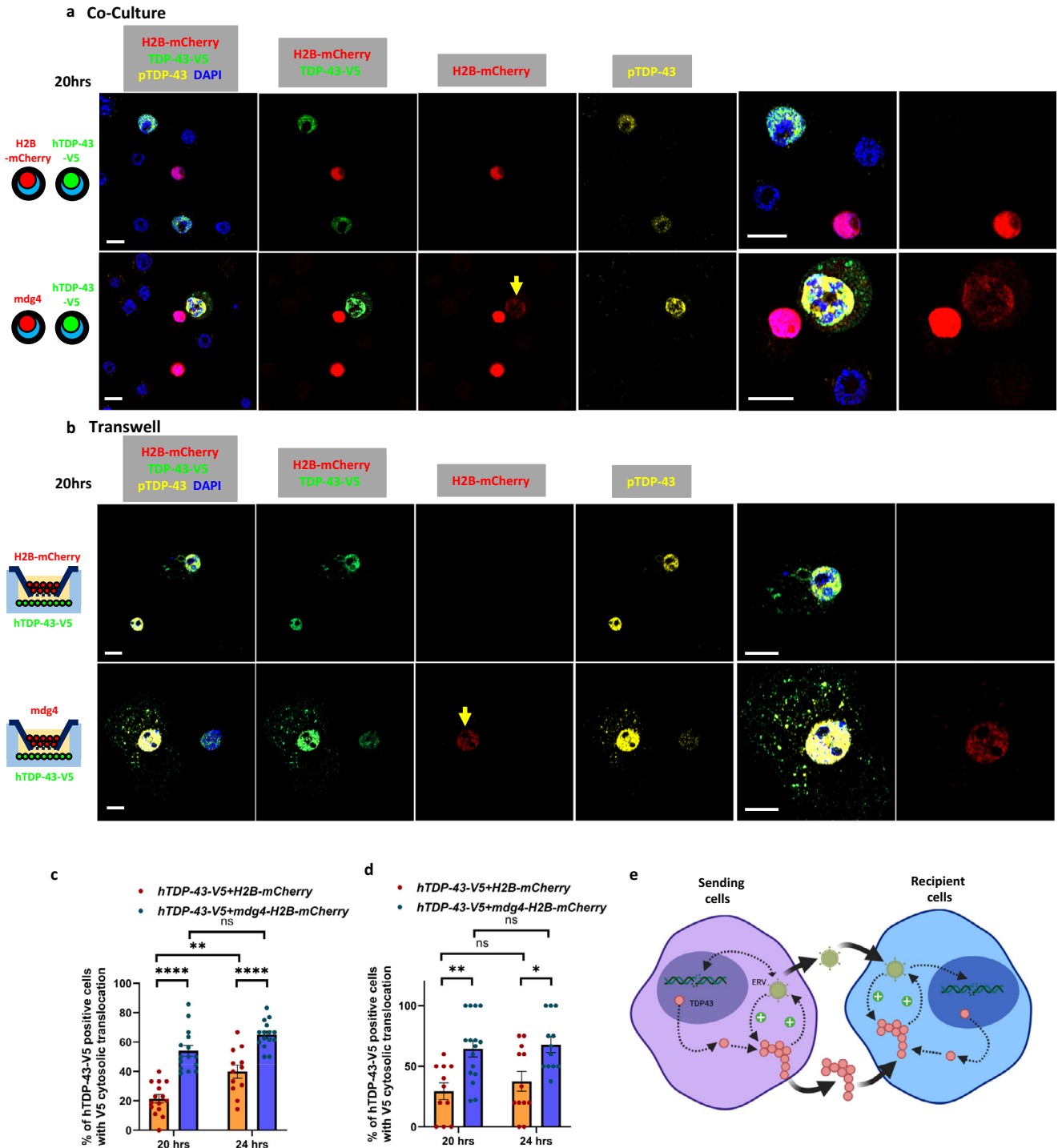

**Fig. 5 | Viral transmission of mdg4-ERV from producing cells is able to cause human TDP-43 proteinopathy in recipient cells. a** Immunofluorescent images of S2 cells with staining for the V5 tag on TDP-43-V5 (green), pTDP-43 (yellow), H2B-mCherry (red) and DAPI (blue). H2B-mCherry and mdg4-H2B-mCherry as the control and producing cells were co-cultured with TDP-43-V5 recipient cells. Scale bar = 5 μm. **b** Immunofluorescent images of S2 cells with staining for V5 tag on TDP-43-V5 (green), pTDP-43 (yellow), H2B-mCherry (red) and DAPI (blue). H2B-mCherry (control) and mdg4-H2B-mCherry (mdg4 producing cells) were cultured in opposition to human TDP-43-V5 expressing recipient cells in a transwell system. Scale bar = 5 μm. **c** Quantification of the % of S2 cells with cytoplasmic V5 translocation to cytoplasm in co-culture of hTDP-43-V5 with H2B-mCherry and hTDP-43-V5 with mdg4-H2B-mCherry after 20 h (21.46 ± 2.80%, $n = 15$ examined fields; 54.00 ± 3.65%, $n = 15$ examined fields) and 24 h (39.98 ± 4.45%, $n = 12$ examined fields; 65.07 ± 2.39%, $n = 15$ examined fields) after induction of hTDP-43-V5. Data are

shown as mean ± SEM and a two-way ANOVA with Tukey's multiple comparisons was performed. **$p < 0.01$; ****$p < 0.0001$; ns, no significant difference. **d** Quantification of % of S2 cells with cytoplasmic V5 translocation in the transwell assays of hTDP-43-V5 with H2B-mCherry and human TDP-43-V5 with mdg4-H2B-mCherry after 20 h (29.39 ± 6.81%, $n = 11$ examined fields; 64.31 ± 6.69%, $n = 16$ examined fields) and 24 h (37.50 ± 8.13%, $n = 12$ examined fields; 67.85 ± 6.43%, $n = 12$ examined fields) induction of hTDP-43-V5. Data are shown as mean ± SEM and a two-way ANOVA with Tukey's multiple comparisons was performed. *$p < 0.05$; **$p < 0.01$; ns, no significant difference. **e** Model of feedback amplification between ERV expression and TDP-43 pathology. Intercellular transmission of either TDP-43 aggregates as reported by other groups or of the ERVs, as reported here, is sufficient to mediate re-initiation of the TDP-43 and ERV amplification in recipient cells. *P*-values for significant comparisons are provided in the Source Data file (made in ©BioRender–biorender.com).

feedback amplification to sustain the proteinopathy in other disorders as well. This suggests the potential of clinical interventions to block RTE/ERV expression or function, as a means to interrupt a positive feedback mechanism driving disease progression.

## Methods

### Data reporting

No statistical methods were used to predetermine sample size. Animal age for each experiment was noted throughout the text. The experiments were not randomized, and the investigators were not blinded to allocation during experiments and outcome assessment.

### Fly strains and husbandry

We obtained SPG-Gal4 (R54C07-Gal4, #50472) and UAS-myr-RFP (#7119) from Bloomington *Drosophila* Stock Center. The hTDP-43-KI[28] and UAS-hTDP-43-RFP[39] were generous gifts from Professor David Morton (Oregon Health & Science University) and Professor Jane Y. Wu (Northwestern University School of Medicine). The following stocks, UAS-GFP-IR, UAS-mdg4-IR (also known as UAS-gypsy-IR in previous studies), UAS-hTDP-43, tub-Gal80[ts] and UAS-mCD8-GFP were used in our previous studies[12,14,42,43]. To prevent artifacts from genetic variation between groups, all strains used in this study were backcrossed to our laboratory wild-type strain, Canton-S derivative w[1118] (*isoCJ1*), for at least five generations. Male flies were chosen as the experimental subjects throughout the study. All flies for TARGET[66] temperature-shift experiments (SPG-*Gal4* combined with *tub-Gal80[ts]*) were raised in a 21 °C incubator from the embryonic stage[66]. All flies for TARGET experiments were immediately shifted to the permissive temperature at 29 °C after eclosion, which was designated as day 0. These TARGET flies were then incubated at 29 °C for lifespan assays or dissected at desired time points for analyzing the patterns of molecular markers dependent on the experimental designs.

### *Drosophila* S2 expression constructs

To generate the inducible hTDP-43-V5 construct, the full length of hTDP-43 was PCR amplified but the stop codon was removed in order to generate an in frame fusion protein with the V5 tag provided by the pMT-V5-6xHis plasmid (Invitrogen, V412020). To generate the pMT-H2B-BFP-P2A-hTDP-43-V5 construct, the H2B and BFP were separately PCR amplified from pAc-H2B-mCherry[40] and moxBFP (Addgene, 68064) constructs and then fused by PCR to generate the H2B-BFP fragment. The P2A was separately added to the C-terminal of H2B-BFP and N-terminal of hTDP-43 by PCR. The final H2B-BFP-P2A-hTDP-43 was then created by PCR fusing H2B-BFP-P2A and P2A-hTDP-43 fragments and then cloned into the inducible pMT-V5-6xHis plasmid. The pAc-H2B-mCherry-HA, pAc-mdg4-H2B-mCherry-HA and pAc-mdg4-H2B-mCherry-HA[ΔEnv] constructs were generated described previously[40].

### Mammalian SH-SY5Y expression constructs

To generate the constitutively expressing pcDNA3.1-H2B-mCherry-HA, the H2B-mCherry-HA was moved from pAc-H2B-mCherry-HA by restriction enzyme NotI and reinserted the H2B-mCherry-HA DNA fragment into the NotI site of pcDNA3.1 (Thermo Fisher Scientific, V79020). The backbone of HERV-K (pcDNA3.1-HERV-K) was obtained from Professor Nath's laboratory[13] and used as the template to generate the constitutive pcDNA3.1-HERV-K-P2A-H2B-mCherry-HA plasmid. In short, the P2A-H2B-mCherry-HA fragment was PCR amplified and integrated downstream of Env of pcDNA3.1-HERV-K while removing stop codon of Env.

### Adult fly survival assays

Adult male flies with desired genotypes raised at 21 °C were collected immediately after eclosion and incubated in a 29 °C incubator for survival assay as addressed in previous studies[12,14]. In short, ten adult flies of a given genotype were housed within one vial, and >100 flies in total for experimental group were used for each experiment. The surviving flies from each vial were flipped into fresh vials with fly food every other day, and dead flies at specific time points were recorded for the final survival curve analysis. The Log-Rank (Mantel-Cox) test and the Gehan-Breslow-Wilcoxon test were used to compare the significance of survival curves.

### Immunostaining of adult fly brains

Adult brains at specific time points for each experiment were dissected, and immunofluorescent staining was performed as previously described[12]. Adult brains were dissected in ice-cold phosphate-buffered-saline (PBS) and then transferred into 4% paraformaldehyde (Electron Microscopy Sciences #15713) PBS solution (1XPBS with 4% paraformaldehyde and 0.2% Triton-X-100 (SIGMA-ALDRICH)) and incubated for 30 min under vacuum twice. After fixation, dissected brains were washed three times for 10 min with 1XPBST wash solution (1XPBS with 1% Triton-X-100 and 3% NaCl). Brain samples were then incubated in blocking solution (1XPBST with 10% normal horse serum) overnight at 4 °C on a nutator. After blocking, dissected brains were transferred into primary antibodies and incubated overnight at 4 °C with the following dilutions: mouse anti-Repo (1:10, Developmental Studies Hybridoma Bank 8D12), rat anti-Elav (1:10, Developmental Studies Hybridoma Bank 7E8A10), mouse anti-Elav (1:10, Developmental Studies Hybridoma Bank 9F8A9), mouse anti-Flag (1:100, SIGMA-ALDRICH F3165), rabbit anti-pTDP-43 (1:100, SIGMA-ALDRICH SAB4200223), rabbit anti-pTDP-43 (1:100, proteintech 22309-1-AP) and rabbit anti-γH2Av (1:150, Rockland Immunochemicals 600-401-914) in 1XPBST washing solution with 10% normal horse serum. Samples were then washed with 1XPBST four times, 15 min each. Fly brains were incubated in secondary antibody solution (1XPBST washing solution with 10% normal horse serum) at 4 °C overnight. Secondary antibodies were combined from the following lists: donkey anti-mouse Alexa Fluor 488 (Jackson ImmunoResearch Laboratories, 715-545-151), donkey anti-rabbit Alexa Fluor 488 (Jackson ImmunoResearch Laboratories, 711-545-152), donkey anti-rat Alexa Fluor 594 (Jackson ImmunoResearch Laboratories, 712-585-153), donkey anti-mouse Alexa Fluor 647 (Jackson ImmunoResearch Laboratories, 715-605-151), donkey anti-rat DyLight 405 (Jackson ImmunoResearch Laboratories, 712-475-153). After washing with 1XPBST for 4 × 15 min, fly brains were mounted in FocusClear (CelExplorer, FC-101), imaged using a Zeiss LSM 800 with Airyscan mode, and acquired images were processed by the Zeiss Zen software package.

### Quantification of spreading and severity of TDP-43 pathology and DNA damage in the adult fly brain

To standardize the brain region for comparing changes of molecular markers over time and between different genetic groups, patterns of molecular markers in a defined 5595.7825 μm$^2$ (77.99 μm x 71.75 μm) area (delineated as the red dashed rectangle shown in Fig. 2a) One corner of the selected area (red dashed rectangle) was set at the overlap between the central brain and optic lobe. This area of each brain then was quantified. In order to monitor the spread of pTDP-43 or γH2Av, the farthest cell within the same row relative to the SPG membrane margin with positive molecular markers was given a number dependent on its sequential cell position away from the SPG membrane within its row (as shown in the Fig. 2i, Supplementary Fig. 3b). All numbers assigned to each row in the selected brain region at a particular time window were averaged to calculate a "spreading score" of that genotype at that time-point. The "impact score" was designed to compare the fraction of cells labeled with a given marker (pTDP-43 or γH2Av) in a selected genotype and time point. In short, the neuronal cells with pTDP-43 or γH2Av within the selected brain region were counted and divided by the total number of Elav$^+$ neurons in that region to calculate a percentage of neuronal cells labeled.

## Quantification of neuronal loss in the adult fly brain

For assessment of TDP-43 pathology induced in the SPG cell effecting on bystander neurons, the change of total neuronal number in the same selected red dash rectangle (as shown in Fig. 2a) from different genotypes over time were counted. Neurons were marked with Elav staining, and then the total Elav⁺ neurons in the red dash rectangle was measured.

## Cell culture and transfection conditions

*Drosophila* S2 cells (Thermo Fisher Scientific, R69007) were maintained in Schneider's Drosophila medium (Thermo Fisher Scientific, 21720001) supplemented with 10% fetal bovine serum (FBS) (Thermo Fisher Scientific, 10438026) and 100U/ml Penicillin-Streptomycin (Thermo Fisher Scientific, 15140122) in a 25 °C incubator. Human neuroblastoma cells, SH-SY5Y (ATCC, CRL-2266), were grown in Dulbecco's Modified Eagle Medium/Nutrient Mixture F-12 (DMEM/F-12) (Thermo Fisher Scientific, 11320033) with 10% FBS and 100U/ml Penicillin-Streptomycin in a 37 °C incubator with 5% $CO_2$ supplementation.

Conditions for S2 cells transfection were modified from our previous studies[40,42]. $5 \times 10^5$ cells were seeded onto coverslips coated with 0.5 mg/ml concanavalin A (ConA) (MP Biomedicals, 195283) and placed in the 6 well culture plate overnight prior to the transfection. Single or co-transfections with desired construct combinations (pAc-H2B-mCherry, pAc-mdg4-P2A-H2B-mCherry and pMT-hTDP-43-V5) were performed with 1 μg of DNA each and Effectene transfection reagents (Qiagen, 301427). Post 24 h transfection, transfection complexes were washed away and supplied with the fresh complete medium containing 700 μM $CuSO_4$ to induce hTDP-43-V5 expression.

Transfection conditions of SH-SY5Y were as follows: $8 \times 10^5$ cells were seeded onto coverslips coated with 0.1 mg/ml Poly-D-Lysine and placed in 6 well culture plates overnight prior to transfection. Single transfection was performed under the manufacturer's guideline with a mixture of 4 μg pcDNA3.1-H2B-mCherry or pcDNA3.1-HERV-K-P2A-H2B-mCherry with 12 μl TransIT-2020 reagent in a 1:3 ratio (Mirus, MIR 5400). After 24 h of incubation, transfection complexes were washed away, and fresh complete medium was supplied daily until the desired time points for analysis.

## Co-culture and transwell setup for cellular assays of *Drosophila* S2 cells

In order to expand the donor cell population and enhance the production of virus-like particles (VLPs), S2 cells were transfected with pAc-H2B-mCherry or pAc-mdg4-P2A-H2B-mCherry 48 h before co-culture or transwell assays were performed (as flowchart shown in Supplementary Fig. 10). For the hTDP-43-V5 recipient population, the inducible pMT-hTDP-43-V5 was delivered with a 6 h short transfection prior to the experimental setup. For co-culture, $5 \times 10^5$ of donor (pAc-H2B-mCherry or pAc-mdg4-P2A-H2B-mCherry) and recipient (pMT-hTDP-43-V5) cells were seeded at a 1:1 ratio onto the ConA-coated coverslip and maintained for 72 h to enhance the entrance of VLPs into recipient cells. At 20 or 24 h prior to the experimental time, the culture medium was replaced with a medium containing 700 μM $CuSO_4$ to induce hTDP-43-V5 expression in recipient cells. For transwell assays, $5 \times 10^5$ of recipient (pMT-hTDP-43-V5) cells were plated onto ConA-coated coverslips and placed in the bottom well. At the same time, $1 \times 10^6$ of donor (pAc-H2B-mCherry or pAc-mdg4-P2A-H2B-mCherry) cells were placed into the upper well. 20 or 24 h prior to analysis, $CuSO_4$ was added to induce hTDP-43-V5 expression in recipient cells.

## Immunostaining of *Drosophila* S2 and human SH-SY5Y cells

S2 or SH-SY5Y cells on the ConA or Poly-D-Lysine coated coverslips were washed three times with 1XPBS and immediately fixed in 4% paraformaldehyde (Electron Microscopy Sciences #15713) PBS solution (1XPBS with 4% paraformaldehyde and 0.2% Triton-X-100 (SIGMA-ALDRICH)) for 10 min on a nutator. After fixation, cells were washed three times with 1XPBST wash solution (1XPBS with 1% Triton-X-100 and 3% NaCl) and then incubated for 1 h in blocking solution (1XPBST with 10% normal horse serum) at room temperature. Cells were transferred into primary antibodies in 1XPBST washing solution with 10% normal horse serum after blocking and incubated for 1 h at room temperature with the following dilutions: mouse anti-V5 (1:250, Thermo Fisher Scientific R960-25), rabbit anti-TDP-43 (1:200, proteintech 10782-2-AP) and rabbit anti-pTDP-43 (1:200, SIGMA-ALDRICH SAB4200223), rabbit anti-pTDP-43 (1:500, proteintech 22309-1-AP), mouse anti-HERV-K-Env (1:500, AUSTRAL Biologicals HERM-1821-5), rat anti-mCherry (1:500, Thermo Fisher Scientific M11217) and rabbit anti-STMN2 (1:500, Novus NBP1-49461). Samples were then washed with 1XPBST three times for 10 min each. Cells were incubated within secondary antibody solution (1XPBST washing solution with 10% normal horse serum) with DAPI at room temperature for 1 h. After washing three times with 1XPBST for 10 min each, cells were mounted in Pro-Long Diamond Antifade Mountant (Thermo Fisher Scientific), imaged using a Zeiss LSM 800 with Airyscan mode, and acquired images were processed by Zeiss Zen software package.

## RNA preparation from SH-SY5Y cells and RT-qPCR assays of TDP-43 target genes

Total RNA was extracted from pcDNA3.1-H2B-mCherry or pcDNA3.1-HERV-K-P2A-H2B-mCherry transfected SH-SY5Y cells using Trizol reagent (Thermo Fisher Scientific, 15596026) at different time points, and the total RNA extractions were treated with RQ1 DNase (Promega, M6101) before performing reverse transcription. 1 μg of total RNA from each sample was used to synthesize cDNA with SuperScript IV VILO Master Mix (Thermo Fisher Scientific, 11756050). qPCR was then performed using PowerUp SYBR Green master mix (Thermo Fisher Scientific, A25741) on an Applied Biosystems QuantStudio 3 Real Time PCR System. The primer sets against POLDIP3α (Forward: 5'-CCAAAACCATCCAGGTTCCACAGCAG-3' and Reverse: 5'-GTGGTGGAGAAAGCCGCCTGAG-3') and GAPDH (Forward: 5'-GTTCGACAGTCAGCCGCATC-3' and Reverse: 5'-GGAATTTGCCATGGGTGGA-3') used in this study were as previously described[25]. To obtain the relative fold changes of POLDIP3α, values from each group were further normalized to the levels in pcDNA3.1-H2B-mCherry transfection at Day 2 in order to obtain the relative levels.

## Sequential fractionation of cellular TDP-43 aggregates for Western Blot analysis

The soluble and insoluble TDP-43 fractions within the S2 cells were sequentially extracted dependent on the solubilities in the extraction buffers as previously described[27]. Transfected S2 cells were washed, collected with 1XPBS, and then homogenized in ice-cold RIPA buffer (Thermo Fisher Scientific, 89900) with protease inhibitors (Thermo Fisher Scientific, A32965) and phosphatase inhibitors (Roche, 4906845001). Cell lysates were centrifuged at 16000 g for 30 min at 4 °C, and the supernatant was saved as the RIPA-soluble fraction. The RIPA-insoluble pellets were washed with RIPA buffer by centrifuging at 16000 g for 15 min at 4 °C. Supernatants were discarded and the pellets were extracted and homogenized with urea buffer (7 M Urea, 4% CHAPS, 2 M Thiourea, 30 mM Tris, pH = 8.5). The lysates were centrifuged at 16000 g for 30 min at 4 °C. The supernatant was kept as the RIPA-insoluble and urea-soluble fraction.

30 μg protein samples from each cellular fraction were separated by 10% polyacrylamide gel (BIO-RAD, 4561034) and followed by a canonical Western Blot protocol. In short, membranes were blocked in 1XTris Buffered Saline-Tween-20 (TBST) (1XTBS + 0.2% Tween-20) with 5% SlimFast (chocolate flavor) at room temperature for 30 min. After blocking, membranes were incubated with primary antibodies mouse anti-GFP (1:1000, Thermo Fisher Scientific MA5-15256), and rabbit anti-pTDP-43 (1:1000, proteintech 22309-1-AP) in 5% SlimFast -TBST for 1 h at room temperature or overnight at 4 °C. After washing three times with 1XTBST, membranes were incubated with secondary antibodies

(goat anti-mouse-HRP (Jackson ImmunoResearch Laboratories, 115-035-174) or goat anti-rabbit-HRP (Jackson ImmunoResearch Laboratories, 111-035-144) at 1:1000 or 1:2000 dilutions) for 1 h at room temperature. Signals were developed in chemiluminescent HRP substrate (Millipore-Sigma, WBKLS0100) and imaged with Sapphire Biomolecular Imager (Azure Biosystems).

## Statistics and reproducibility

Statistical analyses for different experimental setups were performed using GraphPad Prism 9. Specific statistical analyses used for comparing the significance between experimental groups were addressed in the corresponding figure legends and the experimental reproducibility was also listed in the figure legends. The significant levels were indicated as star numbers in the following format: $*p < 0.05$, $**p < 0.01$, $***p < 0.001$ and $****p < 0.0001$.

## Reporting summary

Further information on research design is available in the Nature Portfolio Reporting Summary linked to this article.

## Data availability

There are no large datasets that would require deposition into public repositories. All data are reported in the manuscript, and in the source data file, with the exception of multiple independent confocal images. But we place no restrictions on data availability and will arrange to make raw confocal images available upon request. Source Data are available as a Source Data file. Source data are provided with this paper.

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

## Acknowledgements

We thank Professor Avindra Nath and Dr. Wenxiu Li (National Institutes Neurological Diseases and Stroke) for providing the HERV-K backbone. We also thank Professor David Morton (Oregon Health & Science University) and Professor Jane Y. Wu (Northwestern University School of Medicine) sending us the hTDP-43^{WT-KI} and UAS-hTDP-43-RFP transgenic strains. We thank Roger Sher, Wanhe Li, Grigori Enikolopov, Maria de la Paz Fernandez and Tim Mosca for comments on the manuscript, and Enas Gad Elkarim, Lillian Talbot, Sarah Krupp, Meng-Fu Shih and Jorge Aspurua for helpful discussions. This work was supported by NIA awards RF1AG057338, R01AG078788 and RF1AG076493 to J.D.

## Author contributions

Y.-H.C. and J.D. designed the experiments. Y.-H.C. performed all of the experiments and collected and analyzed all the data. Y.-H.C. and J.D. wrote the manuscript.

## Competing interests

The authors declare no competing interests.
