## [Peer Review File · Nature Communications]

Endogenous Retroviruses and TDP-43 Proteinopathy Form a Sustaining Feedback Driving Intercellular Spread of Drosophila NeurodegenerationREVIEWER COMMENTS

Reviewer #1 (Remarks to the Author):

Chang & Dubnau explore the relationship between TDP-43 expression, endogenous retrovirus (ERV) activity and cellular toxicity, in human and fly experimental models. They conclude that intercellular transmission of mdg4 ERV viral-like particles in fly brain is responsible for TDP-43 cytoplasmic aggregation, and that this is accelerated by a positive feedback loop between TDP-43 and mdg4 (or HERV-K in human cells). The key data here are i) cytoplasmic accumulation of phosphorylated/pathological TDP-43 induced by HERV-K or mdg4 expression reports in human and fly cells, respectively, ii) observation of intercellular propagation of human TDP-43-induced proteinopathy and toxicity from subperineurial glia (SPG) to neighbouring (and more distal) neurons, iii) greatly reduced toxicity in this model upon mdg4 knockdown and iv) proof of mdg4 intercellular transmission and, interestingly, not transmission of TDP-43.

This fascinating study points strongly to ERV-encoded envelope proteins being central to TDP-43-driven pathology. Positive feedback between TDP-43 and ERV expression, in hand with ERV intercellular transmission, provides an elegant explanation for the rapid progression of pathology in ALS and other related neurodegenerative conditions. I found this very interesting to consider, and liked the engaging way the manuscript was written. The data in Figure 1 makes it quite possible that the findings for mdg4 would carry over to HERV-K (and indeed IAPs in mice).

I do not have any major criticisms to offer. The experiments were designed carefully and caveats noted where applicable. I do have many questions, and can see a lot of future scope for additional experiments, for example using a non-SPG initiating focus in fly, or CRISPRa to upregulate HERV-K in human cells, but these would be unreasonable to require here. The conclusions are supported by the presented experiments, in my view.

Minor points:

1) Figure 5E indicates intercellular transmission of TDP-43 aggregates, which has been shown elsewhere but is not found here. In the Discussion, it could be explained further why this was not observed here (from the SPG at least). What is the authors' position?

2) mdg4 knockdown counteracted neuronal loss, did it also increase median lifespan?

3) The Discussion could incorporate a few more sentences relating to follow up; what do the authors see as the most important next step(s)? And what does this mean, if anything, for ALS/FTD therapeutics? A further impetus to inhibit HERV-K? This could be flagged as speculation, or omitted at the authors' discretion, but I thought there probably is cause to talk more directly about what the present findings mean in this regard.

Geoff Faulkner (University of Queensland)

Reviewer #2 (Remarks to the Author):

In this manuscript by Chang and Dubnau, the authors explore a new way to think about what drives TDP-43 pathology, and in turn the impact of TDP-43 pathology on neuronal loss. TDP-43 pathology is the hallmark of neurodegenerative diseases ALS and FTD (and also prominent in AD). TDP-43 pathology consists of nuclear depletion of endogenous TDP-43 and cytoplasmic accumulation of TDP-43 inclusions that are phosphorylated. It has remained a challenge for the field to recapitulate both aspects of pathology. The authors use a previously generated transgenic Drosophila line that has the fly TDP-43 gene replaced with an epitope-tagged human TDP-43 cDNA. They demonstrate that

these flies are normal and viable but that they exhibit robust TDP-43 pathology and loss of nuclear TDP-43 upon focal upregulation of hTDP-43.

The authors have previously demonstrated that TDP-43 functions to repress an endogenous retrovirus (e.g., Chang and Dubnau *Current Biology* 2019; Li et al., *PLoS One* 2012; Krug et al., *PLoS Genet*).

This new work seems very similar to the concepts and results presented in the 2019 Chang and Dubnau paper (PMID: 31495585), with a few new advances:

1) Expressing the human retrovirus HERV-K in neuroblastoma cells is sufficient to induce robust phosphorylated TDP-43 pathology.

2) Demonstration that TDP-43 nuclear clearance occurs upon triggering TDP-43 upregulation focally.

3) RNAi knockdown of endogenous retrovirus *mdg4* is sufficient to mitigate the spread of TDP-43 pathology and neuronal loss in their fly model. This result provides evidence that the spread of this retrovirus is responsible for the cell non autonomous effects of TDP-43 expressed in a subset of glia on neurons at a distance.

Overall, the new results provide further evidence for a potential connection between TDP-43 and endogenous retroviruses and support a model whereby TDP-43 functions to repress the expression of the viruses, but TDP-43 pathological changes cause these to be de-repressed and to spread to nearby glia and neurons causing neuronal loss. Likewise, the viruses are able to induce further TDP-43 pathology (e.g., pTDP-43).

In my opinion, this current work and results seem too similar with the 2019 paper to represent a major conceptual advance, although I note above what seem like new developments. I suggest that the authors consider more clearly emphasizing the major advances of this work over their recent paper. I also have several other suggestions for the authors to consider:

1) The loss of nuclear TDP-43 in their fly model is intriguing and represents a potential opportunity for the authors to define which TDP-43 targets are dysregulated upon *mdg4* spread.

2) Is the mechanism of *mdg4* entry to fly cells known? It seems like if the authors could prevent *mdg4* from entering adjacent cells (by knocking down a virus entry factor) that could allow them to provide direct evidence if virus entry is responsible for inducing TDP-43 pathology and subsequent neuron loss at a distance.

3) In Fig. 1 the authors robustly induce TDP-43 pathology (cytoplasmic pTDP-43) by expressing HERV-K. Is this treatment sufficient to cause nuclear TDP-43 clearance as well? Because this is a human system the authors could consider testing if any of TDP-43's known splicing targets (e.g., *STMN2* or *POLDIP3*) are dysregulated upon HERV-K expression.

Reviewer #3 (Remarks to the Author):

This manuscript from Yung-Heng Chang and Josh Dubnau provides an interesting and provocative new hypothesis implicating endogenous retroviruses (ERVs) as mediators of TDP-43-induced spread of neurodegeneration. The authors utilize cultured human and *Drosophila* cells as well as in vivo experiments in existing and newly-developed *Drosophila* models of TDP-43 proteopathy. The authors provide convincing evidence that TDP-43 induces ERV expression as previously reported, and that induced ERV expression drives the appearance of TDP-43 pathology and associated toxicity in neighboring cells.

Interestingly, the authors find that ERV-induced TDP-43 pathology in neighboring cells is not due to the physical transfer of TDP-43 between the donor and recipient cell. Overall, the findings of this manuscript are novel and, for the most part, supported by the experimental evidence provided. The major issue with the manuscript is an overly complex and difficult to digest presentation. It requires a lot of work on the part of the reader to figure out what is being depicted in the figures. Data presentation is very confusing and should be simplified/restructured to allow readers to properly appreciate the findings. In its current form, the rather exciting findings of this manuscript risk the possibility of being overlooked and underappreciated by anyone who is not a *Drosophila* geneticist with expertise in *Drosophila* models of neurodegeneration. While the issues included in point 5 below may each seem minor on their own, they ultimately amount to a major issue that, if unfixed, will limit the impact of the study.

Major issues:

1. **Abstract:** "We also demonstrate that viral ERV transmission causes propagation of such TDP-43 pathology to cells..." sets the reader up to expect that ERVs mediate the prion-like spread of TDP-43. It should be made clear in the abstract that ERV-induced TDP-53 pathology in neighboring cells is not due to the physical transfer of pTDP-43 between the donor and recipient cell. This is a very interesting and unexpected finding that should be emphasized from the beginning.
2. **Results:** The authors state "Taken together, these results indicate that viral transmission of the mdg4-ERV triggers the appearance of TDP-43 protein pathology in recipient cells." This statement leads the reader to believe that ERV-derived viral particles mediate the appearance of pathological TDP-43 in neighboring neurons. The authors have not actually shown that ERV-derived viral particles form in this model and drive the appearance of TDP-43 in neighboring neurons. This should be tested experimentally or specifically mentioned as a caveat in the Discussion.
3. There is lack of clarity on how calculations were performed for each quantification. The authors do not sufficiently explain their methods for calculating spreading score, impact score, neuron counting (what does "neuron number" mean in Fig. 2i?), and spreading row. These calculations need to be thoroughly explained in the Methods. The data is difficult to critically assess without this information.
4. Figures are quite difficult to interpret. This is a major issue throughout the manuscript. These are just a few examples:
 - a. Fig. 1a: The cartoon makes it appear that the Env domain is not a part of HERV-K.
 - b. Fig. 1c: What are the dots and bar at the 100% mark? Are these data points? Throughout the figures, data points should be sized so that readers can visualize each data point.
 - c. Fig. 2c-i: The authors should label what the Flag antibody detects (hTDP-43) rather than writing "Flag" on the top label.
 - d. Fig. 3b-e: The authors should denote what GFP and RFP actually represent in the figures rather than labeling them "GFP" and "RFP." GFP should be changed to "mCDS" and RFP to "hTDP-43."
 - e. Fig. 5a, b: With current labeling, it is difficult to immediately understand that the top row is recipient cells transfected with control pAc-H2B-mCherry and the bottom row is recipient cells transfected with pAc-mdg4-H2B-mCherry. These labels (and all others) should be simplified.
 - f. Fig. 5: Panel c precedes panel b.
 - g. Black backgrounds of image labels make them difficult to read.
 - h. All bar graphs are too small in relation to the images.
 - i. Data presentation should be consistent throughout (i.e. not a mix of bar graphs and box-and-whisker).

Response to reviewer comments on NCOMMS-22-30966-T

We were pleased to receive such constructive reviewer comments, and a very positive decision from the editors. Most of the reviewer comments were already quite supportive of the previous submitted version of our manuscript, and many of the critiques were ones that we were able to answer with edits to the manuscript text and figures. But there were several cases where adequately addressing the reviewer comments required that we conduct new experiments. The revised manuscript includes a series of new experiments that we feel should satisfy each of the reviewer critiques. Here we provide a point by point response to each critique in which we outline how we have addressed the reviewer concern (either edits to the text or figures, or addition of new data).

Reviewer #1 (Remarks to the Author):

Chang & Dubnau explore the relationship between TDP-43 expression, endogenous retrovirus (ERV) activity and cellular toxicity, in human and fly experimental models. They conclude that intercellular transmission of mdg4 ERV viral-like particles in fly brain is responsible for TDP-43 cytoplasmic aggregation, and that this is accelerated by a positive feedback loop between TDP-43 and mdg4 (or HERV-K in human cells). The key data here are i) cytoplasmic accumulation of phosphorylated/pathological TDP-43 induced by HERV-K or mdg4 expression reports in human and fly cells, respectively, ii) observation of intercellular propagation of human TDP-43-induced proteinopathy and toxicity from subperineurial glia (SPG) to neighbouring (and more distal) neurons, iii) greatly reduced toxicity in this model upon mdg4 knockdown and iv) proof of mdg4 intercellular transmission and, interestingly, not transmission of TDP-43.

This fascinating study points strongly to ERV-encoded envelope proteins being central to TDP-43-driven pathology. Positive feedback between TDP-43 and ERV expression, in hand with ERV intercellular transmission, provides an elegant explanation for the rapid progression of pathology in ALS and other related neurodegenerative conditions. I found this very interesting to consider, and liked the engaging way the manuscript was written. The data in Figure 1 makes it quite possible that the findings for mdg4 would carry over to HERV-K (and indeed IAPs in mice).

I do not have any major criticisms to offer. The experiments were designed carefully and caveats noted where applicable. I do have many questions, and can see a lot of future scope for additional experiments, for example using a non-SPG initiating focus in fly, or CRISPRa to upregulate HERV-K in human cells, but these would be unreasonable to require here. The conclusions are supported by the presented experiments, in my view.

Minor points:

1) Figure 5E indicates intercellular transmission of TDP-43 aggregates, which has been shown elsewhere but is not found here. In the Discussion, it could be explained further why this was not observed here (from the SPG at least). What is the authors' position?"

The reviewer raises an interesting point. We cannot be sure whether there may be some inter-cellular movement of TDP-43 aggregates. We can only state that we do not observe evidence of this. It is possible that there is some movement that is below the level at which we are able to detect it. What we can say though is that there is appearance of TDP-43 pathological inclusions in recipient cells (neurons in this case), and that this is driven by the mdg-4 ERV. It may be that movement of the ERV and of TDP-43 protein act together. We did attempt to write the discussion so that it is clear that we are not ruling out movement of TDP-43 protein, but rather we are presenting a new (potentially additional) mechanism of inter-cellular spread of pathology.

“2) mdg4 knockdown counteracted neuronal loss, did it also increase median lifespan?”

Great question! We have added a new figure panel (Supplementary Fig. 9h) to address this. We used the same RNAi transgene to knock down mdg4 expression in the SPG, and find that this also extends lifespan of the animals.

“3) The Discussion could incorporate a few more sentences relating to follow up; what do the authors see as the most important next step(s)? And what does this mean, if anything, for ALS/FTD therapeutics? A further impetus to inhibit HERV-K? This could be flagged as speculation, or omitted at the authors' discretion, but I thought there probably is cause to talk more directly about what the present findings mean in this regard.”

Thanks for this suggestion. Our discussion section previously offered this speculation: “But our findings raise the possibility that RTE and ERV expression provides a feedback amplification to sustain the proteinopathy in other disorders as well.” To address this reviewer request, we have added this sentence: “This suggests the potential of clinical interventions to block RTE/ERV expression or function, as a means to interrupt a positive feedback mechanism driving disease progression.”

Geoff Faulkner (University of Queensland)

Reviewer #2 (Remarks to the Author):

In this manuscript by Chang and Dubnau, the authors explore a new way to think about what drives TDP-43 pathology, and in turn the impact of TDP-43 pathology on neuronal loss. TDP-43 pathology is the hallmark of neurodegenerative diseases ALS and FTD (and also prominent in AD). TDP-43 pathology consists of nuclear depletion of endogenous TDP-43 and cytoplasmic accumulation of TDP-43 inclusions that are phosphorylated. It has remained a challenge for the field to recapitulate both aspects of pathology.

The authors use a previously generated transgenic *Drosophila* line that has the fly TDP-43 gene replaced with an epitope-tagged human TDP-43 cDNA. They demonstrate that these flies are normal and viable but that they exhibit robust TDP-43 pathology and loss of nuclear TDP-43 upon focal upregulation of hTDP-43.

The authors have previously demonstrated that TDP-43 functions to repress an endogenous retrovirus

(e.g., Chang and Dubnau *Current Biology* 2019; Li et al., *PLoS One* 2012; Krug et al., *PLoS Genet*).

“This new work seems very similar to the concepts and results presented in the 2019 Chang and Dubnau paper (PMID: 31495585), with a few new advances:

1) Expressing the human retrovirus HERV-K in neuroblastoma cells is sufficient to induce robust phosphorylated TDP-43 pathology.

2) Demonstration that TDP-43 nuclear clearance occurs upon triggering TDP-43 upregulation focally.

3) RNAi knockdown of endogenous retrovirus mdg4 is sufficient to mitigate the spread of TDP-43 pathology and neuronal loss in their fly model. This result provides evidence that the spread of this retrovirus is responsible for the cell non autonomous effects of TDP-43 expressed in a subset of glia on neurons at a distance.

Overall, the new results provide further evidence for a potential connection between TDP-43 and endogenous retroviruses and support a model whereby TDP-43 functions to repress the expression of the viruses, but TDP-43 pathological changes cause these to be de-repressed and to spread to nearby glia and neurons causing neuronal loss. Likewise, the viruses are able to induce further TDP-43 pathology (e.g., pTDP-43).

In my opinion, this current work and results seem too similar with the 2019 paper to represent a major conceptual advance, although I note above what seem like new developments.”

The reviewer correctly notes that we have previously published a study that demonstrates that expression of TDP43 in *Drosophila* glia drives non-cell autonomous toxicity to neurons, and that this requires expression in the glia of the mdg4 ERV. And the reviewer correctly notes that our current manuscript breaks new ground by demonstrating that HERV-K and mdg4 can induce TDP-43 pathology, that HERV-K over-expression drives nuclear clearance, and that knockdown of mdg4 is sufficient to mitigate the spread of pathology and nuclear loss. But we hope we can convince the reviewer that there are additional new findings in this manuscript that really distinguish it from prior work. Notably, all of our prior work on retrotransposons and ERVs, indeed all of the literature, focus on activation of these elements as downstream effectors of TDP43 pathology. Here, we demonstrate for the first time that ERV expression is not only driven by TDP-43 pathology, it is a driver of TDP-43 pathology. That is a totally new concept, and it raises the prospect of a feedback amplification, also new. In addition, this is the first dataset that supports the idea that ERVs can move between cells, and then and initiate TDP-43 pathology. So, it provides a new mechanism to explain the inter-cellular propagation of such pathology. We feel, and hope the reviewer agrees, that these are significant advances over our previous publications.

“I suggest that the authors consider more clearly emphasizing the major advances of this work over their recent paper. “

We thank the reviewer for this suggestion. We have revised the discussion section with this goal in mind. Our revised discussion now includes the following text:

“We also have previously demonstrated in a *Drosophila* model that RTE/ERV expression mediates many of the neurodegenerative effects of TDP-43 dysfunction^{12,14}. Here, we demonstrate that human HERV-K or the *Drosophila* mdg4-ERV also are upstream drivers of TDP-43 pathology, and are sufficient to trigger dysfunctional phosphorylation, cytoplasmic translocation, and formation of insoluble forms of TDP-43. Thus, TDP-43 pathology stimulates ERV expression^{12-14,16,18} and ERV expression triggers pathological changes in TDP-43. We also demonstrate that inter-cellular viral transmission of mdg4 is sufficient to re-initiate TDP-43 pathology in recipient cells. And this feedback mechanism underlies propagation of both TDP-43 pathology and neurodegeneration in a *Drosophila* in vivo context.

“I also have several other suggestions for the authors to consider:

1) The loss of nuclear TDP-43 in their fly model is intriguing and represents a potential opportunity for the authors to define which TDP-43 targets are dysregulated upon mdg4 spread.

2) Is the mechanism of mdg4 entry to fly cells known? It seems like if the authors could prevent mdg4 from entering adjacent cells (by knocking down a virus entry factor) that could allow them to provide direct evidence if virus entry is responsible for inducing TDP-43 pathology and subsequent neuron loss at a distance.

3) In Fig. 1 the authors robustly induce TDP-43 pathology (cytoplasmic pTDP-43) by expressing HERV-K. Is this treatment sufficient to cause nuclear TDP-43 clearance as well? Because this is a human system the authors could consider testing if any of TDP-43’s known splicing targets (e.g., STMN2 or POLDIP3) are dysregulated upon HERV-K expression.”

We have addressed each of the above 3 comments by addition of new data that we feel should satisfy the reviewer. The new data are as follows:

Point 1: We agree with the reviewer that the observed nuclear clearance of TDP-43 suggests a loss of function that should be reflected in splicing/expression of known targets. There are well established targets of TDP-43 in mammalian systems, but less so in *Drosophila*. So we chose to address this issue in our human Neuroblastoma culture system (see point 3).

Point 2: This is also an interesting and important question. While cellular virus entry factors are not well characterized, the viral Env glycoprotein is known to be required to mediate entry into recipient cells. We tested the effects of an mdg4-H2B-mCherry construct in which the Env gene was deleted (mdg4-H2B-mCherry^{ΔEnv}). This Env deleted virus can be expressed and even replicate within a cell (as we demonstrated previously, Keegan et al., PLoS Genetics, 2019), but is unable to transmit between cells. We demonstrate using our transwell system (revised Supplementary Fig. 10c,d), that this Env deleted virus is unable to promote TDP-43 pathology in cells grown across the transwell membrane. This is in contrast with the effects that we document for the mdg-4 virus that has an intact Env gene.

Point 3: We show new evidence HERV-K expression in neuroblastoma cells also is associated with loss of nuclear TDP-43 (revised Supplementary Fig. 1d) and results in the predicted mis-regulation of STMN2 and POLDIP3 α , two established targets (revised Supplementary Fig. 1a-c).

Reviewer #3 (Remarks to the Author):

This manuscript from Yung-Heng Chang and Josh Dubnau provides an interesting and provocative new hypothesis implicating endogenous retroviruses (ERVs) as mediators of TDP-43-induced spread of neurodegeneration. The authors utilize cultured human and *Drosophila* cells as well as in vivo experiments in existing and newly-developed *Drosophila* models of TDP-43 proteopathy. The authors provide convincing evidence that TDP-43 induces ERV expression as previously reported, and that induced ERV expression drives the appearance of TDP-43 pathology and associated toxicity in neighboring cells. Interestingly, the authors find that ERV-induced TDP-43 pathology in neighboring cells is not due to the physical transfer of TDP-43 between the donor and recipient cell. Overall, the findings of this manuscript are novel and, for the most part, supported by the experimental evidence provided.

“The major issue with the manuscript is an overly complex and difficult to digest presentation. It requires a lot of work on the part of the reader to figure out what is being depicted in the figures. Data presentation is very confusing and should be simplified/restructured to allow readers to properly appreciate the findings. In its current form, the rather exciting findings of this manuscript risk the possibility of being overlooked and underappreciated by anyone who is not a *Drosophila* geneticist with expertise in *Drosophila* models of neurodegeneration. While the issues included in point 5 below may each seem minor on their own, they ultimately amount to a major issue that, if unfixed, will limit the impact of the study. “

We thank the reviewer for this very helpful and constructive comment. We have attempted to address each of the issues as listed below, and also have tried to improve the readability of the manuscript and figures, and have added some discussion points requested by other reviewers that we hope also will help.

“Major issues:

1. Abstract: *“We also demonstrate that viral ERV transmission causes propagation of such TDP-43 pathology to cells...” sets the reader up to expect that ERVs mediate the prion-like spread of TDP-43. It should be made clear in the abstract that ERV-induced TDP-43 pathology in neighboring cells is not due to the physical transfer of pTDP-43 between the donor and recipient cell. This is a very interesting and unexpected finding that should be emphasized from the beginning.”*

We have revised the abstract accordingly. It now states “We also demonstrate that viral ERV transmission causes initiation of such TDP-43 pathology in recipient cells that express physiological levels of TDP-43, whether they are in contact or at a distance.”

“2. Results: The authors state “Taken together, these results indicate that viral transmission of the mdg4-ERV triggers the appearance of TDP-43 protein pathology in recipient cells.” This statement leads the reader to believe that ERV-derived viral particles mediate the appearance of pathological TDP-43 in neighboring neurons. The authors have not actually shown that ERV-derived viral particles form in this model and drive the appearance of TDP-43 in neighboring neurons. This should be tested experimentally or specifically mentioned as a caveat in the Discussion.”

This is an excellent point. We have addressed this in two ways. First, we now provide experimental evidence that ENV glycoprotein mediated viral transmission is required to drive the appearance of TDP-43 pathology in recipient cells grown in culture. But we also nuance our discussion of this issue so that it is clear that the fact that propagation of TDP-43 pathology is mediated by viral transmission is only shown in cell culture. We have edited the relevant section of the discussion so that it now reads:

“We also demonstrate that inter-cellular viral transmission of mdg4 is sufficient to re-initiate TDP-43 pathology in recipient cells grown in culture. The ERV-TDP-43 feedback mechanism underlies propagation of both TDP-43 pathology and neurodegeneration in a *Drosophila* in vivo context, consistent with the idea that viral transmission also is at play.”

“3. There is lack of clarity on how calculations were performed for each quantification. The authors do not sufficiently explain their methods for calculating spreading score, impact score, neuron counting (what does “neuron number” mean in Fig. 2i?), and spreading row. These calculations need to be thoroughly explained in the Methods. The data is difficult to critically assess without this information.”

We have extensively edited the methods section so that it more clearly describes how these spreading and impact scores were calculated. This description now reads:

“To standardize the brain region for comparing changes of molecular markers over time and between different genetic groups, patterns of molecular markers were analyzed in a defined $5595.7825\mu\text{m}^2$ ($77.99\mu\text{m} \times 71.75\mu\text{m}$) area (delineated as the red dashed

rectangle shown in Fig. 2a). One corner of the selected area (red dashed rectangle) was set at the overlap between the central brain and optic lobe. Signal in this area of each brain then was quantified. In order to monitor the spread of pTDP-43 or γ H2Av, each cell nucleus was assigned a row number according to the number of cell diameters distance from the SPG membrane at the brain surface to that nucleus (as shown in the Fig. 2i, Supplementary Fig. 3b). The 'row numbers' assigned to labelled nucleus in the selected brain region were averaged to calculate a "spreading score" of that genotype at that time-point. The "impact score" was designed to compare the fraction of cells labeled with a given marker (pTDP-43 or γ H2Av) in the selected brain region for each genotype and time point. In short, the neuronal cells with pTDP-43 or γ H2Av within the selected brain region were counted and divided by the total number of Elav⁺ neurons in that region to calculate a percentage of neuronal cells labelled."

In addition, to address the reviewer's point that we had not fully described what 'neuron number' means in Fig. 2i, we added this description to the method's section:

"For assessment of the effects of SPG-induced TDP-43 pathology on nearby neurons, we quantified the total number of labelled neuronal nuclei in the same selected red dash rectangle (as shown in Fig. 2a). The neuronal nuclei were marked with the Elav antibody, and then the total number of Elav⁺ neurons in the red dashed rectangle was counted for each brain."

"4. Figures are quite difficult to interpret. This is a major issue throughout the manuscript. These are just a few examples:

a. Fig. 1a: The cartoon makes it appear that the Env domain is not a part of HERV-K. "

Thanks, we have changed the coloring so that it should be visually clear that Env is part of HERV-K.

"b. Fig. 1c: What are the dots and bar at the 100% mark? Are these data points? Throughout the figures, data points should be sized so that readers can visualize each data point."

In our revised figures, we show all data points wherever possible. In this particular bar graph (Fig. 1c), we have now reduced the size of the symbols for each data point so that individual data points are visible for the HERV-K group. There are cases in some figures where the sample size is too large for all individual points to be seen, even when we reduce the size of the data point symbol because some of the data points pile on top of each other.

"c. Fig. 2c-i: The authors should label what the Flag antibody detects (hTDP-43) rather than writing "Flag" on the top label. "

Done both in this figure, and the supplementary figures.

d. Fig. 3b-e: The authors should denote what GFP and RFP actually represent in the figures rather than labeling them “GFP” and “RFP.” GFP should be changed to “mCD8” and RFP to “hTDP-43.”

In our revised figures, we have done exactly as requested, except that instead of labeling RFP-hTDP-43 as “TDP-43” we have labeled it fully as “RFP-hTDP-43”.

e. Fig. 5a, b: With current labeling, it is difficult to immediately understand that the top row is recipient cells transfected with control pAc-H2B-mCherry and the bottom row is recipient cells transfected with pAc-mdg4-H2B-mCherry. These labels (and all others) should be simplified.

Thanks for this constructive critique. In our revised Fig. 5 we have attempted to greatly simplify these labels by use of cartoons that should clarify this to the reader.

f. Fig. 5: Panel c precedes panel b.

Fixed.

g. Black backgrounds of image labels make them difficult to read.

Thanks, we have changed the backgrounds to grey, which makes lettering easier to read.

h. All bar graphs are too small in relation to the images.

We have somewhat reduced the sizes of the images to provide space to increase the size of the bar graphs. We hope this is sufficient. There is not space in these multi panel images to make the bar graphs even larger. If the editors and reviewers feel it is best, we could move the bar graphs to the supplement, but that would make it more challenging for a reader to easily see quantification next to the representative images.

i. Data presentation should be consistent throughout (i.e. not a mix of bar graphs and box-and-whisker).

Done.

REVIEWERS' COMMENTS

Reviewer #1 (Remarks to the Author):

The authors have addressed my comments, thank you. The new longevity data in Supplementary Fig. 9h are very interesting.

Geoff Faulkner (University of Queensland)

Reviewer #2 (Remarks to the Author):

The authors have revised their manuscript and have addressed my previous concerns, especially highlighting the major advances of their new findings over the previous literature about retrotransposons and TDP-43. Their new findings show that these can function BOTH downstream and upstream of TDP-43 pathology is very intriguing. Their experiments connecting mdg4 to STMN2 cryptic splicing is very exciting and further connects mdg4 to TDP-43 pathology. I recommend publication of this revised manuscript in its present form.

Reviewer #3 (Remarks to the Author):

My comments have been addressed - congratulations on a very interesting manuscript.

Response to reviewers:

We thank the reviewers for very thoughtful and constructive critiques, and we are pleased that all three reviewers feel that we have fully addressed those critiques.